**JCB** Journal of Cell Biology

# Acidosis attenuates the hypoxic stabilization of HIF-1α by activating lysosomal degradation

Bobby White[1], Zhenyi Wang[1]*, Matthew Dean[1]*, Johanna Michl[1], Natalia Nieora[1], Sarah Flannery[2], Iolanda Vendrell[2], Roman Fischer[2], Alzbeta Hulikova[1], and Pawel Swietach[1]

**Hypoxia-inducible factors (HIFs) mediate cellular responses to low oxygen, notably enhanced fermentation that acidifies poorly perfused tissues and may eventually become more damaging than adaptive. How pH feeds back on hypoxic signaling is unclear but critical to investigate because acidosis and hypoxia are mechanistically coupled in diffusion-limited settings, such as tumors. Here, we examined the pH sensitivity of hypoxic signaling in colorectal cancer cells that can survive acidosis. HIF-1α stabilization under acidotic hypoxia was transient, declining over 48 h. Proteomic analyses identified responses that followed HIF-1α, including canonical HIF targets (e.g., CA9, PDK1), but these did not reflect a proteome-wide downregulation. Enrichment analyses suggested a role for lysosomal degradation. Indeed, HIF-1α destabilization was blocked by inactivating lysosomes, but not proteasome inhibitors. Acidic hypoxia stimulated lysosomal activity and autophagy via mammalian target of rapamycin complex I (mTORC1), resulting in HIF-1α degradation. This response protects cells from excessive acidification by unchecked fermentation. Thus, alkaline conditions are permissive for at least some aspects of HIF-1α signaling.**

## Introduction

Tumor hypoxia arises from an elevated respiratory demand pitched against inadequate oxygen supply from dysfunctional vascular perfusion (Thomlinson and Gray, 1955; Vaupel et al., 1989) and places constraints on metabolism, hence proliferation. Consequently, transcriptional responses to hypoxia underpin a fundamental survival adaptation (Schödel and Ratcliffe, 2019). Many of these responses are mediated through hypoxia-inducible factors (HIFs) and transcription regulators stabilized at low $O_2$ partial pressure ($pO_2$) (Jaakkola et al., 2001; Maxwell et al., 1999; Semenza, 2000). The presence of HIF signaling in every nucleated human cell hints to a major and seemingly unconditional role in transducing hypoxia (Pugh and Ratcliffe, 2017; Wang and Semenza, 1993). However, hypoxic signaling must be contextualized to conditions experienced by tissues. For example, in vivo hypoxia is mechanistically linked to acidosis because both relate to inadequate perfusion (Rohani et al., 2019; Vaupel et al., 1989). Indeed, the diffusion barrier between cells and capillaries that maintains hypoxia gradients similarly restricts washout of $CO_2$ and lactic acid, the acidic products of respiration and fermentation (Swietach et al., 2023). Tumor hypoxia is expected alongside acidosis, albeit it in a stoichiometry defined by metabolic profile (Vaupel et al., 1989), and both chemical features should be considered in studies of HIF signaling.

A widely practiced method for imposing hypoxia in vitro is to initiate incubation at reduced $pO_2$, starting from alkaline medium conditions. Such experimental protocols cause $pO_2$ to fall before medium begins to acidify, leading to an artefactual uncoupling between $pO_2$ and extracellular pH (pHe). This maneuver removes any potential interaction between hypoxia and acidity on HIF signaling and does not replicate the development of acidotic hypoxia in the tumor microenvironment (TME) (Rohani et al., 2019). The interplay between hypoxia and acidosis on HIF signaling is unclear because evidence has been conflicting (Filatova et al., 2016; Mekhail et al., 2004; Parks et al., 2013; Selfridge et al., 2016; Tang et al., 2012; Willam et al., 2006). A number of reports have studied the interaction between acidosis and HIF (Parks et al., 2013; Selfridge et al., 2016; Tang et al., 2012; Willam et al., 2006), although some noted a discordance between the effect on HIF proteins and their targets (Willam et al., 2006). One study demonstrated acid-stimulated nucleolar sequestration of the Von Hippel–Lindau tumor suppressor (pVHL), which mediates proteasomal degradation of HIF-α subunits (Mekhail et al., 2004). Others suggested a pVHL-independent synergy between acidosis and HIF, implicating heat shock protein-90 instead (Filatova et al., 2016).

Efforts to investigate the role of pH in hypoxic signaling must ensure adequate control and physiological relevance. However,

[1]Department of Physiology, Anatomy and Genetics, University of Oxford, Oxford, UK;  [2]Target Discovery Institute, Centre for Medicines Discovery, Nuffield Department of Medicine, University of Oxford, Oxford, UK.

*Z. Wang and M. Dean contributed equally to this paper.   Correspondence to Pawel Swietach: pawel.swietach@dpag.ox.ac.uk.



a common experimental intervention to manipulate medium pH is supplementation with nonvolatile buffers (e.g., HEPES) followed by titration outside an incubator (Filatova et al., 2016; Mekhail et al., 2004; Parks et al., 2013; Tang et al., 2012). This approach demonstrably leads to unstable pH under subsequent $CO_2$ incubation (Michl et al., 2019). The recommended approach to controlling pH is by varying the $CO_2$ partial pressure ($pCO_2$)/[$HCO_3^-$] ratio, in accordance with the Henderson–Hasselbalch equation. In the under-perfused setting of acidic tumors, the extent to which $pCO_2$ rises and [$HCO_3^-$] falls will depend on their diffusivity. Compared with $CO_2$ gas, which penetrates both intra- and extracellular spaces, $HCO_3^-$ has a slower diffusion coefficient and can only travel contiguously through the smaller extracellular space of tissues (Swietach et al., 2023). Thus, tumor acidosis is more likely to take the form of metabolic acidosis (reduced [$HCO_3^-$]) than respiratory acidosis (raised $CO_2$). While a notable study on HIF imposed acidosis by raising incubation $pCO_2$ (Selfridge et al., 2016), metabolic acidosis has not been investigated. Another consideration is the choice of cell line for studying pHe/pO2 interplay (Willam et al., 2006). Many cell lines survive poorly at low pHe (Michl et al., 2024), which could give erroneous interpretations of the actions of pHe on hypoxic signaling. Thus, it is prudent to select acid-resistant cell lines that can support viable signaling under the harsh combination of acidosis and hypoxia. To that end, we sought to measure the effects of controlled metabolic acidosis on HIF responses to hypoxia among the more acid-resistant colorectal cancer (CRC) cell lines.

We hypothesize that acidosis feeds critical information for contextualizing the effect of hypoxia on HIF signaling. In general, HIF responses could be considered a means of augmenting oxygen delivery and reducing consumption, ostensibly to limit the extent of hypoxia (a deleterious chemical stress), but these actions can also affect pH. Responses such as angiogenesis (Liu et al., 1995) also facilitate the removal of acid, whereas others, like the switch from respiration to fermentation (Firth et al., 1994; Kim et al., 2006), have the opposite effect on tumor pHe. Excessive acidification of the TME is problematic for fermentative phenotypes because low pH acutely blocks glycolysis (Blaszczak et al., 2022; Bock and Frieden, 1976). Thus, embarking on complete HIF signaling in an acidic environment can inadvertently block cells from harnessing energy and resources by transcriptionally inactivating respiration and allowing low pH to block fermentation. Our results indicate that hypoxic stabilization of HIF-1α is subservient to pHe, a protective effect that develops gradually through the stimulation of lysosomal HIF-1α degradation.

## Results

### Identifying acid-resistant CRC cell lines most likely to manifest acidosis/hypoxia interplay

To study the impact of acidosis on hypoxic responses, we first identified suitable CRC lines that are likely to demonstrate a meaningful interaction between acidosis and hypoxia on HIF signaling, that is, survive adequately at low pHe and have the metabolic capacity to reduce pO2 and pHe. Previously, we ranked 68 CRC cell lines by their acid sensitivity, measured in terms of cell survival after 6-day culture over a range of starting medium pH set by varying [$HCO_3^-$] (Michl et al., 2024). These datasets, supplemented with additional replicates, highlighted eight lines that span a phenotypic spectrum from most acid-sensitive (COLO320DM and COLO678) to most acid-resistant (SW1222 and C99) (Fig. 1 A). Next, we measured the capacity of these CRC lines to reduce pHe and pO2 by performing fluorometric assays (Fig. 1, B and C). In these experiments, low buffering allows metabolic acid production to detectably change medium pH, whereas the oil barrier allows respiration to deplete medium O2 without rapid atmospheric O2 ingress (Blaszczak et al., 2024). Of the cell lines tested, acid-resistant C99 and SW1222 were notable for reducing pHe and pO2 concurrently (Fig. 1 B), even after 48 h pre-treatment at pHe 6.4 (Fig. 1 C). To account for the effect of low pHe on growth, seeding density was adjusted to give comparable cell numbers between the pHe 7.4 and pHe 6.4 pre-treatment groups, verified by Cell Tracker Orange (CTO) measurements, except for most acid-sensitive COLO320DM, which survived poorly at low pHe (Fig. S1 A). Thus, C99 and SW1222 cells were selected for subsequent experiments because their acid-resistant phenotype is expected to produce acidotic hypoxia, akin to solid tumors (Rohani et al., 2019). To test for generalization of our findings, additional experiments used HT29 or HDC111 as representing intermediate phenotypes.

### Responses to acidosis and HIF expose distinct metabolic vulnerabilities

Respiration and fermentation are blocked, respectively, by hypoxia and acidosis; therefore, feedback circuits are warranted to titrate the appropriate balance between these metabolic pathways. This interplay was inferred from time courses of medium acidification and oxygen depletion. Whereas the latter is an intuitive surrogate of respiration, both fermentative lactic acid and respiratory $CO_2$ production may contribute to medium acidification. We previously showed the former (nonvolatile) acid to be the major contributor to medium acidification (Blaszczak et al., 2022). Confirming this, we repeated metabolic profiling in HT29, HDC111, C99, and SW1222 cells, terminating experiments at 8 or 17 h and relating cumulative $H^+$ production with end point (lactate). These metrics correlated 1:1 (Fig. 1 D), indicating that our assay measures fermentative rate. Next, metabolic profiling of C99 cells was performed after 48 h pre-treatment in alkaline (pHe 7.4) or acidic (pHe 6.4) media with or without dimethyloxalylglycine (DMOG), a prolyl hydroxylase inhibitor that stabilizes HIF-α. Seeding densities were adjusted to yield comparable cell numbers at the point of measurements (Fig. S1 B). Profiling was performed in assay media titrated to the pre-treatment pHe but without DMOG, reasoning that any transcriptional response to HIF is likely maintained in the measurement window. Cumulative $H^+$ production and O2 consumption time courses are shown in Fig. 1 E. DMOG treatment at pHe 7.4 shifted the metabolic phenotype from respiratory to fermentative, in line with the canonical HIF response that diverts metabolic flows. However, low pHe suppressed fermentation, even after DMOG treatment. Concurrently, DMOG-triggered respiratory suppression persisted

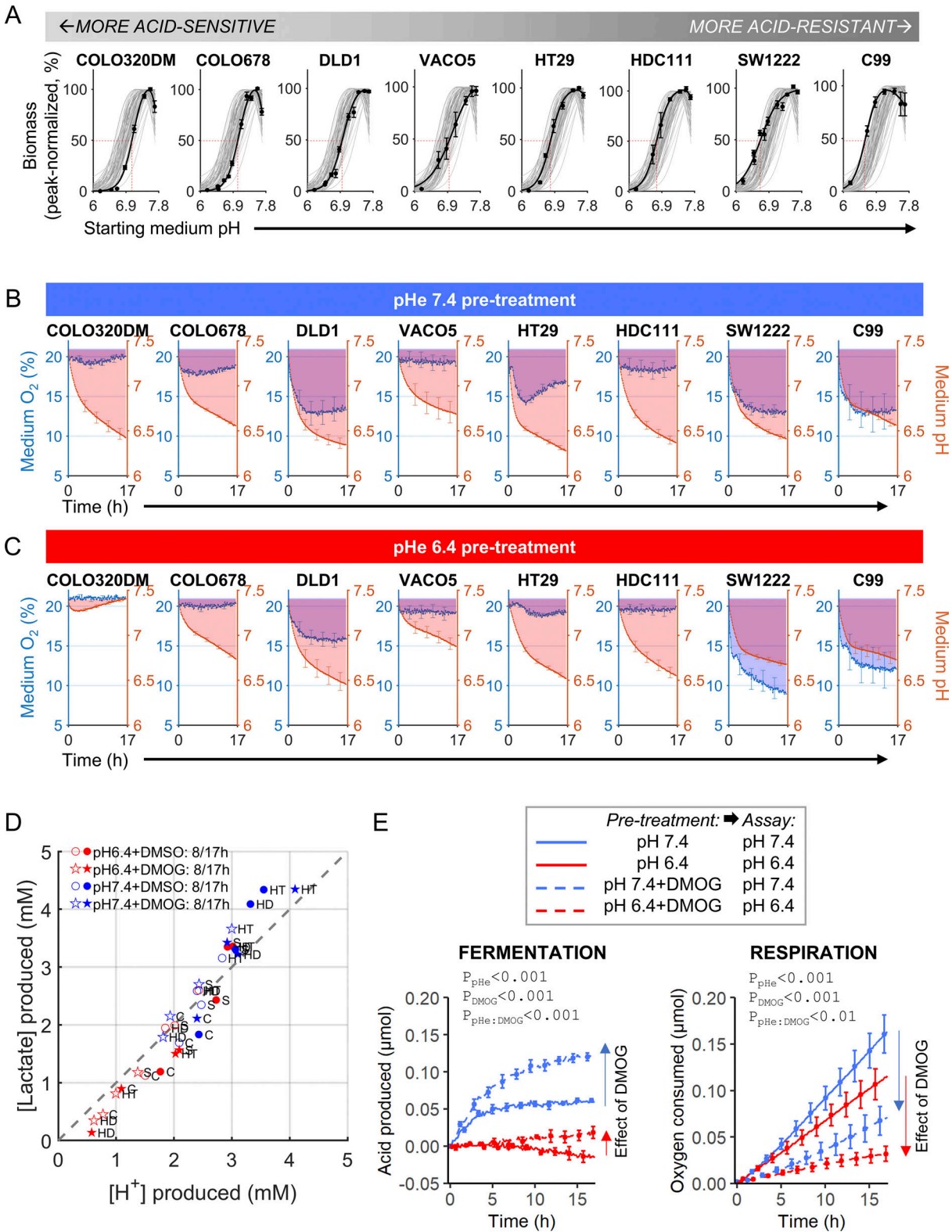

Figure 1. **Phenotyping CRC cell lines for survival and metabolic responses to pHe changes. (A)** Eight CRC cell lines were selected from a panel of 68 lines to span a range from acid-sensitive (left) to acid-resistant (right). Cells were cultured for 6 days in media set to a starting pH ranging from 6.2 to 7.7. At endpoint, cellular biomass was measured by SRB assay and normalized to the maximal growth measured for each repeat for at least three independent repeats. Data fitted to a biphasic Hill-type survival curve (black), contextualized against previously measured survival curves for 68 CRC cell lines (grey). **(B)** CRC cell lines were pre-treated in alkaline (pH 7.4) media for 48 h, prior to fluorimetric measurements of medium acidification and oxygenation, from a starting condition of normoxia and pHe 7.4. **(C)** Measurements repeated on cells pre-treated in acidic (pH 6.4) media for 48 h. Following acidic pre-treatment, conditions were returned to pHe 7.4 and normoxia immediately prior to commencing fluorimetric measurements. **(D)** Paired [lactate] and [H+] measurements.

C99 (C), SW1222 (S), HDC111 (HD), or HT29 (HT) cells were pre-treated for 48 h in alkaline (pH 7.4) or acidic (pH 6.4) media containing either DMOG (1 mM) or its vehicle (DMSO). After treatment, metabolic profiling was performed in DMOG-free medium from a starting condition of normoxia and pHe 7.4. After either 8 h (empty symbol) or 17 h (filled symbol), media samples were collected for [lactate] assays. **(E)** Metabolic profiling of C99 cells for acid production and oxygen consumption. Cells were pre-treated for 48 h in alkaline (pH 7.4) or acidic (pH 6.4) media containing either DMOG (1 mM) or its vehicle (DMSO). After pre-treatment, metabolic profiling was performed under DMOG-free conditions, but pHe remained unchanged. Cumulative acid production and oxygen consumption were recorded simultaneously as readouts of fermentation and respiration, respectively. Experiments performed for at least three independent repeats. Data shown as mean ± SEM. Statistical testing by three-way ANOVA (see Table S1 for full results).

under acidosis. Overall, acidotic hypoxia produced a state of reduced fermentative and respiratory metabolism, ostensibly to prevent excessive acidification. To explain these findings, we postulate that certain HIF responses are influenced by pH, which we tested by interrogating the hypoxic signaling pathway.

## Maintained HIF-1α stabilization under hypoxia is contingent on alkaline conditions

To describe the possible interactions between hypoxia and pHe on HIF signaling, cells were cultured in one of several formulations containing 0–22 mM $HCO_3^-$, which equilibrate to a pHe in range 6.2–7.4 under 5% $CO_2$ (Michl et al., 2019), followed immediately by hypoxic incubation. This protocol imposes a hypoxic stimulus concurrently with a pHe change that spans a range of possible acid production-to-$O_2$ consumption ratios in vivo (Lyng et al., 1997; Swietach et al., 2023). After 48 h, deemed sufficient to evoke most transcriptional responses, lysates were analyzed for HIF-1α and HIF-2α. In the presence of hypoxia, HIF-1α induction in C99 cells was reduced monotonically as pHe decreased (Fig. 2 A). In contrast, the pHe dependence of hypoxic HIF-2α induction was biphasic, peaking at pHe 6.9 (Fig. 2 B). Since the pHe response was more prominent with HIF-1α, this isoform was selected for further studies. The inhibitory effect of acidosis on HIF-1α stabilization under hypoxia was confirmed in another acid-resistant line, SW1222 (Fig. 2 C), and two lines (HT29 and HDC111) of intermediate acid sensitivity (Fig. 2, D and E), arguing for a more general phenomenon.

In the aforementioned experiments, low pHe was modeled experimentally as a metabolic acidosis, i.e., reduced $[HCO_3^-]$ at constant $CO_2$. It is therefore possible that the observations are attributable to $HCO_3^-$ sensing rather than pHe. However, this alternative was rejected by experiments that reduced $[HCO_3^-]$ alongside $pCO_2$ to maintain pHe at 7.4. Lower $[HCO_3^-]$ at alkaline pHe did not alter hypoxic HIF-1α induction (Fig. 2 F), indicating that the trigger observed in acidic media was the rise in $[H^+]$. In the case of fermentative tumors, the rise in extracellular $[H^+]$ is linked to higher [lactate]; therefore, a more physiologically accurate formulation of acidosis should consider lactate. This was tested by replacing 20 mM NaCl with sodium lactate. HIF-1α stabilization remained suppressed under acidotic hypoxia, relative to alkalotic hypoxia, even when extracellular [lactate] was raised in acidosis (Fig. 2 G). We found no significant interaction between lactate and low pHe, indicating that the $H^+$ ion per se, rather than lactic acid, is responsible for the observed attenuation of HIF signaling.

## Acidosis suppresses hypoxic responses in a subset of HIF targets

The pH dependence of HIF-1α stabilization is expected to manifest as differential abundance of proteins under acidotic hypoxia versus alkalotic hypoxia. This was tested in an unbiased manner using label-free proteomic analyses of SW1222 cells after 48 h treatment in acidosis, hypoxia, or their combination. >9,000 proteins were identified across the samples (Table S2). Responses to acidotic and alkalotic hypoxia resolved well on principal component analysis (Fig. 3 A). Two-way ANOVA with batch and false-discovery corrections revealed statistically significant responses to hypoxia and acidosis, and their interaction (Fig. 3 B). These included 2,302 proteins responding to acidosis, 3,304 responding to hypoxia, and an interaction for 1,259 proteins (Table S3). Consistent with immunoblot results (Fig. 2), HIF-1α abundance increased under hypoxia, but this response was attenuated under acidotic conditions (Fig. 3 C, top row; Fig. 3 D). Protein responses that follow the HIF-1α pattern were discovered by correlation to log$_2$-transformed quantifications (Fig. 3 C). Responses of 59 proteins correlated significantly with HIF-1α (Table S4), including canonical HIF targets such as carbonic anhydrase 9 (CA9) and pyruvate dehydrogenase kinase 1 (PDK1) (Kim et al., 2006) (Fig. 3 D). This finding suggests that the hypoxic induction of at least some HIF targets becomes blunted when hypoxia is paired with acidosis. However, many proteins did not follow the HIF-1α pattern (Fig. 3 D), typically because their hypoxic response became stronger under acidosis (e.g., CEACAM5) or were strongly induced by acidosis but insensitive to hypoxia (e.g., AKR1C2).

Attenuated hypoxic HIF-1α stabilization under acidosis could reflect reduced HIF-1α synthesis and/or increased degradation. To gain insight, we tested for an enrichment of short half-life proteins in the four experimental conditions, reasoning that proteins most prone to breakdown would have decreased abundance under conditions that activate a degradative pathway. Using a list of 508 proteins with half-life <8 h (Li et al., 2021), enrichment analysis (Fisher's exact test) found no significant associations under either acidosis or hypoxia analysed separately, but a highly significant association ($P < 0.0001$) under acidotic hypoxia, specifically among downregulated proteins. This analysis suggests at least some role for a degradative process, activated under acidotic hypoxia, in reducing the abundance of short-lived proteins that include HIF-1α.

Further analyses considered all proteins that are differentially abundant under hypoxia, irrespective of pHe (Fig. 3 E). Clustering identified four response groups: hypoxic upregulation strongly attenuated at low pHe, hypoxic upregulation moderately attenuated at low pHe, hypoxic upregulation strengthened at low pHe, and pHe-independent downregulation by hypoxia (Table S5). Abundance of these proteins, plotted as a scatter diagram, revealed the influence of pHe on hypoxic responses (Fig. 3 F). Here, upregulated proteins falling below the line of identity followed the HIF-1α pattern and included

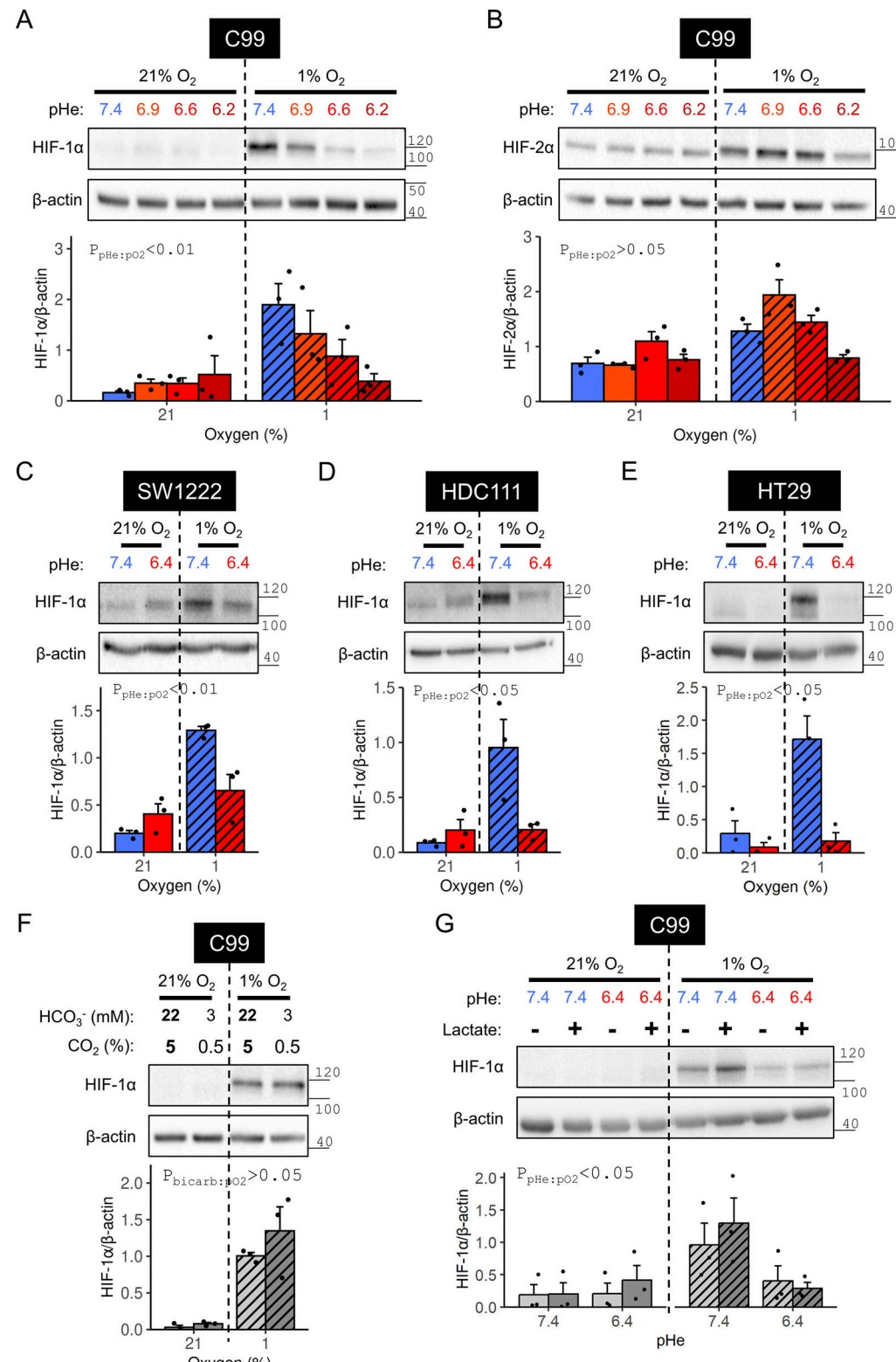

Figure 2. **pHe dependence of hypoxic HIF-α induction in acid-resistant CRC cell lines.** C99, SW1222, HDC111, or HT29 cells were incubated under normoxia (21% $O_2$) or hypoxia (1% $O_2$) in media of pH ranging from 6.2 to 7.4 for 48 h. **(A–E)** After treatment, lysates were analyzed for (A, C, D, and E) HIF-1α or (B) HIF-2α immunoreactivity. **(F)** HIF-1α stabilization under hypoxia is not dependent on $[HCO_3^-]$ at constant pHe. C99 cells were grown for 48 h in media at pH 7.4 containing either 22 mM $HCO_3^-$ in an atmosphere of 5% $CO_2$ or 3 mM $HCO_3^-$ in an atmosphere of 0.5% $CO_2$. During the incubations, cells were exposed to normoxia or hypoxia and lysates were analyzed for HIF-1α immunoreactivity. **(G)** Acidosis impairs hypoxic HIF-1α induction independently of [lactate]. C99 cells were incubated under normoxia or hypoxia in media of pH 7.4 or 6.4 with or without 20 mM lactate for 48 h and analyzed for HIF-1α immunoreactivity. HIF-α signals were normalized to loading control (β-actin) for three independent repeats. Datapoints indicate individual repeats, and bars indicate mean + SEM. Statistical testing by two- or three-way ANOVA (see Table S1 for full results). Source data are available for this figure: SourceData F2.

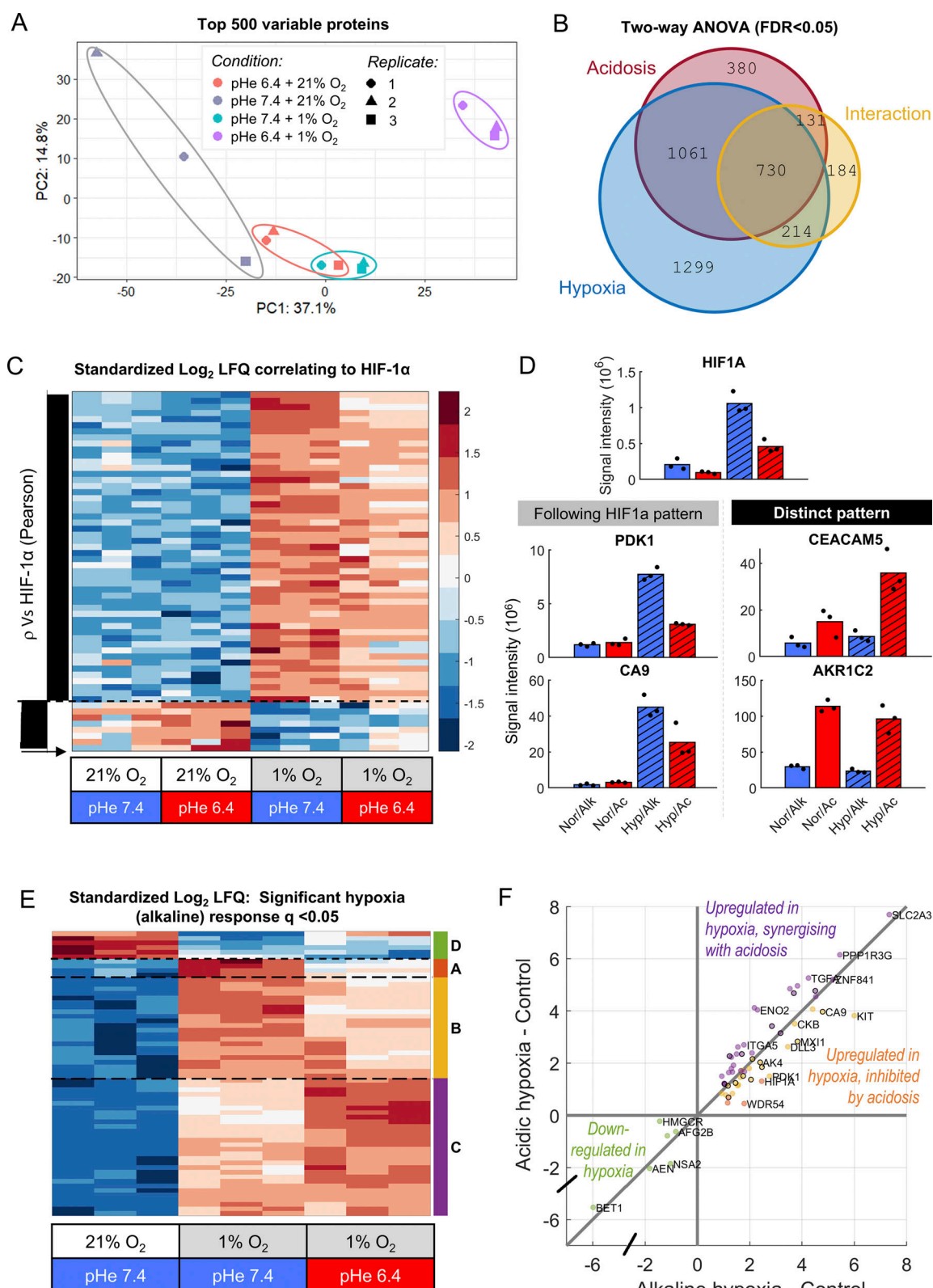

Figure 3. **Proteomic analyses identify protein abundance responses to hypoxia and acidosis.** Label-free mass spectrometry of SW1222 cell lysates after 48 h of culture in media of pH 6.4 or 7.4 under normoxia (21% O₂) or hypoxia (1% O₂). Lysates from three independent replicates. **(A)** Principal component analysis showing the four conditions tested. **(B)** Euler diagram summarizing the results of two-way ANOVA (FDR < 0.05). **(C)** Heatmap of log₂-transformed standardized label-free quantification (LFQ) protein abundance, ranked by significant (FDR < 0.05) Pearson's correlation against HIF-1α (from top row). **(D)** Signal intensities for selected proteins that follow the HIF-1α pattern (CA9, PDK1) and those with distinct responses, notably synergy between acidosis and hypoxia (CEACAM5) and hypoxia-insensitive acid induction (AKR1C2); bars indicate mean, and black datapoints indicate individual replicates. Conditions are

labeled as "Nor" (normoxia), "Hyp" (hypoxia), "Ac" (pHe 6.4), and "Alk" (pHe 7.4). **(E)** Heatmap of log$_2$-transformed standardized LFQ for protein abundance sensitive to alkaline hypoxia (q < 0.05), alongside response in acidic hypoxia. Clustering identifies a downregulated group of proteins "D" (6 proteins: AEN, AFG2B, BET1, ERAL1, HMGCR, and NSA2), an upregulated group "A" that is strongly inhibited under acidosis (4 proteins: HIF1A, KDM4A, SCD, and DR54), an upregulated group "B" that is partially inhibited under acidosis (22 proteins: AK4, ANKZF1, APOD, CA9, CKB, DLL3, EGLN1, GBE1, HMGCS2, IREB2, KDM4B, KDM5B, KIT, LDHA, MXI1, NARF, P4HA1, PDCD4, PDK1, PFKFB4, PHYH, and SORL1), and an upregulated group "C" that synergizes with acidosis (30 proteins: BNIP3L, COL17A1, CORO2A, DPYS, ENO2, GDPD3, GPRC5A, HID1, ITGA5, ITGB6, ITIH3, KDM3A, KDM5C, LRP1, MYO1D, NDRG1, P4HA2, PFKP, PIK3AP1, PLIN2, PLOD2, PPP1R3G, QSOX1, SEMA4B, SLC16A3, SLC2A3, TCAF2, TGFA, UPK2, and ZNF841). **(F)** Scatter plot shows hypoxic response under alkalosis versus acidosis for the four clusters (groups A–D). Black outlines denote canonical HIF targets (from Buffa and Lombardi).

products of cancer-wide conserved HIF-1α target genes (Buffa et al., 2010; Lombardi et al., 2022). Significantly, most products of HIF-1α target genes were less abundant at reduced pH, but this could not be explained by a global rundown of protein levels because many proteins recorded their highest abundance under the combination of acidosis and hypoxia.

To validate the proteomic analyses, immunoreactivity to canonical HIF targets was probed using independently collected C99 and SW1222 lysates. Levels of CA9 and PDK1 exhibited a significant interaction between acidosis and hypoxia (Fig. 4, A and B; and Fig. S2 A). For CA9, this interaction was also observed when HIF-1α was stabilized with DMOG (Fig. 4 C and Fig. S2 B) using DMSO as a vehicle (Fig. S2 C). Furthermore, acidotic hypoxia produced a weaker induction of *CA9* mRNA compared with alkalotic hypoxia (Fig. S2 D). Lactate dehydrogenase A (LDHA) responses correlated less strongly with HIF-1α levels, with modest acid suppression of hypoxic induction in SW1222 cells (Fig. S2 A) but not in C99 cells (Fig. 4 A). Likewise, acidosis attenuated the DMOG response of LDHA in SW1222 (Fig. 4 C) but not C99 cells (Fig. S2 B).

To test for HIF responses among non-metabolic targets, caudal type homeobox 1 (CDX1) was probed. This transcription factor is relevant to colorectal epithelium differentiation and contains a putative hypoxia response element (Choi et al., 2007; Yeung et al., 2011). Under alkaline conditions, hypoxia (Fig. S2 E) or DMOG (Fig. 4 D) induced CDX1. This response was attenuated after *HIF1A* knockdown, implicating a HIF-1α–dependent response (Fig. 4 E). Strikingly, *CDX1* mRNA (Fig. S2 F) and CDX1 protein (Fig. S2 E and Fig. 4 D) were no longer hypoxia- or DMOG-inducible under acidic conditions. Overall, our results demonstrate that HIF-target gene induction becomes attenuated when hypoxia is paired with acidosis, a relevant setting for many solid tumors in vivo.

Immunoblotting for carcinoembryonic antigen-related cell adhesion molecule 6 (CEACAM6) tested for synergy between acidosis and hypoxia as further evidence against a global rundown of proteins under acidotic hypoxia. As with CEACAM5, hypoxia or DMOG (Fig. 4 F) were found to synergize with acidity to augment CEACAM6 levels. This result indicates that the interaction between hypoxia and acidosis described for HIF-1α signaling likely involves a specialized mechanism, rather than a generalized rundown of cellular proteins.

### The attenuation of hypoxic HIF induction under acidosis is protective

Assessing the functional significance of attenuated hypoxic HIF-1α stabilization under acidosis must consider impact on ensemble pathways. To that end, we measured fermentative and respiratory fluxes fluorometrically. Cells were pre-treated for 48 h with DMOG in media at pHe 6.4 or 7.4 and then assayed for metabolic flux under standardized initial conditions deemed permissive for evaluating fermentative and respiratory capacity, i.e., DMOG-free media at pHe 7.4. CTO measurements confirmed comparable cell densities between pre-treatment conditions at the point of metabolic profiling (Fig. S1 C). In C99 cells, fermentative capacity was unaffected by prior exposure to acidosis in the absence of DMOG (Fig. 5 A). However, fermentative capacity was considerably higher after pre-treatment in DMOG-containing alkaline media, compared to DMOG-containing acidic media (Fig. 5 A). This finding indicates that the overall effect of HIF signaling on the fermentative pathway is subservient to pHe. In contrast, the effect of DMOG on respiratory capacity was pHe-insensitive, indicating an outcome that does not strictly follow the pattern of PDK1, a key enzyme regulating pyruvate commitment to the Krebs cycle (Kim et al., 2006). However, multiple factors influence respiratory flux, including the abundance of electron transport chain subunits (Kim et al., 2006). The complex I subunit NDUFS1 was previously established as essential for enabling respiration in acid-stressed cells (Michl et al., 2022; Ugalde et al., 2004), but is not a canonical HIF target because levels were not induced by hypoxia under alkaline conditions (Fig. 5 B). Normoxic acidosis increased NDUFS1 levels, presumably to accommodate the metabolite flux diverted away from acid-inhibited glycolysis (Michl et al., 2022). Strikingly, NDUFS1 levels were reduced when acidosis was combined with hypoxia, ostensibly to offset the consequences of PDK1 downregulation, thereby keeping respiratory flux suppressed under hypoxia, irrespective of pHe.

To investigate a more generalized cellular outcome of the interaction between hypoxia and acidosis, protein biomass was measured as a proxy of proliferation after 6-day culture of moderately acid-sensitive (HT29 and HDC111) and acid-resistant (SW1222 and C99) lines (Fig. 5 C). Measurements were grouped into five experimental blocks, each featuring a medium change after day 2. In the first block (Exp1), cells were cultured at pHe 7.4 and the period of DMOG treatment was varied: (i) absent, (ii) first 2 days, (iii) last 4 days, or (iv) all 6 days. Growth decreased upon activation of hypoxic signaling, proportionally to the number of days in DMOG. Similar conclusions were reached with the second (Exp2) and third (Exp3) blocks performed at pHe 6.9 and 6.4, respectively. The fourth block (Exp4) compared growth in normoxia at pHe 6.9 to growth with DMOG treatment at pHe 7.4, intended to maximally activate HIF-1α. Strikingly, overall growth was not proportional to the number of days in DMOG, with lower-than-expected biomass associated with early DMOG treatment. A similar outcome was noted for the fifth

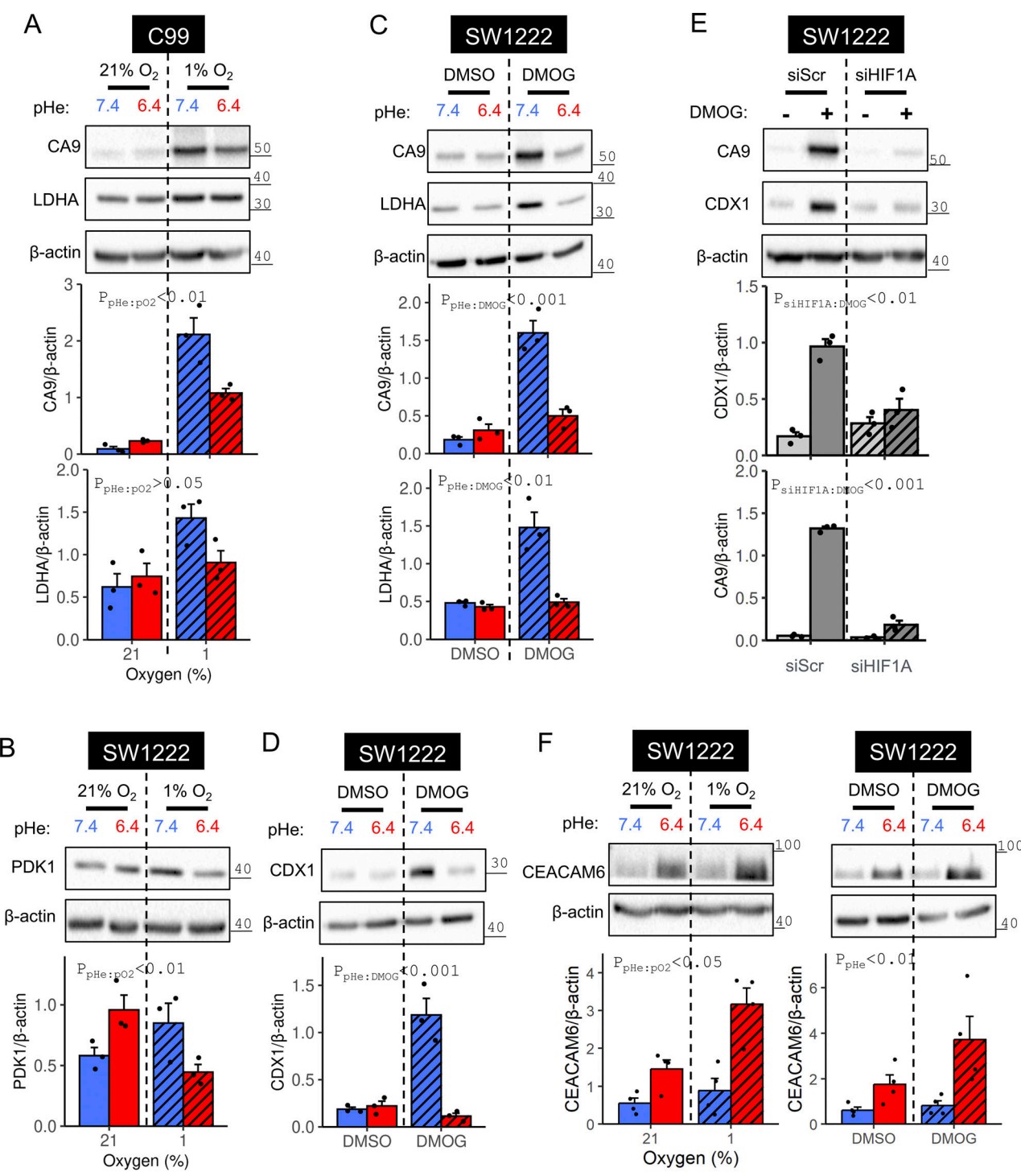

Figure 4. **HIF-target induction under acidotic or alkalotic hypoxia. (A–D)** C99 or SW1222 cells were cultured in media of pH 6.4 or 7.4 and incubated for 48 h in either (A and B) normoxia (21% O₂) or hypoxia (1%), or (C and D) either in the presence of vehicle control (DMSO) or 1 mM DMOG. After treatment, lysates were analyzed for immunoreactivity of canonical HIF targets (CA9, LDHA, PDK1) and the putative HIF target CDX1. **(E)** SW1222 cells were transfected with either non-targeting control siRNA (siScr) or siRNA-targeting *HIF1A* (siHIF1A). 24 h after transfection, cells were cultured for 48 h in media of pH 7.4 containing either DMSO or 1 mM DMOG. Reduced CA9 immunoreactivity in DMOG-treated siHIF1A cells confirmed efficient knockdown. **(F)** Hypoxia synergized with acidosis to strengthen CEACAM6 expression. SW1222 cells were grown in media of pH 6.4 or 7.4 for 48 h. Incubations were performed under either normoxia or hypoxia, or in the presence of either DMSO (vehicle control) or 1 mM DMOG. After treatment, lysates were analyzed for immunoreactivity to CEACAM6. HIF target and CEACAM6 signals were normalized to loading control (β-actin) for three to four independent repeats. Datapoints indicate individual repeats, and bars indicate mean ± SEM. Statistical testing by two-way ANOVA (see Table S1 for full results). Source data are available for this figure: SourceData F4.

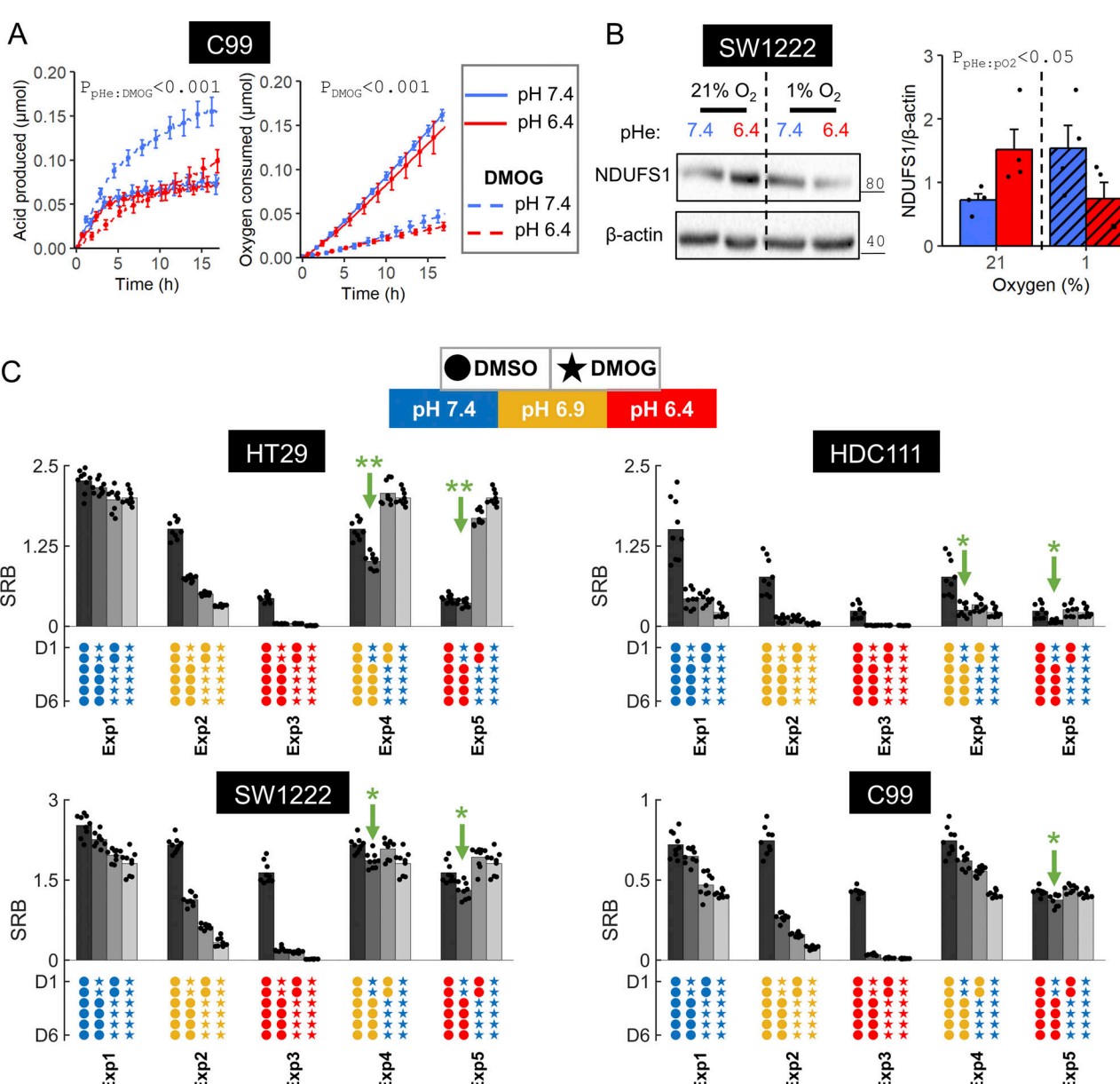

Figure 5. **Effect of the acidosis/hypoxia interaction on metabolic capacity and cell growth. (A)** C99 cells were pre-treated for 48 h with or without 1 mM DMOG in media at pH 6.4 or 7.4. After pre-treatment, metabolic profiling commenced under DMOG-free conditions from a starting pH of 7.4 to assess capacity for fermentation (from acid production) and respiration (from oxygen consumption). **(B)** Complex I downregulation maintains respiratory suppression when hypoxia is imposed concurrently with acidosis. SW1222 cells were cultured in media at either pH 6.4 or 7.4 for 48 h under normoxia (21% $O_2$) or hypoxia (1% $O_2$). After treatment, lysates were analyzed for immunoreactivity to NDUFS1. All immunoblot signals were normalized to loading control (β-actin) for four independent repeats. Note, same loading control used as for Fig 4 B because NDUFS1 and PDK1 were blotted on the same membrane. **(A and B)** Datapoints indicate individual repeats, and bars indicate mean ± SEM. Statistical testing by two- or three-way ANOVA (see Table S1 for full results). **(C)** Cell growth measured in terms of protein biomass (SRB assay) after 6 days of culture, with an intermediary medium change on day 2. Treatment options included DMOG (1 mM) and incubation at pHe 6.4, 6.9, or 7.4. Data are grouped into five experimental blocks of four protocols each. Three SRB measurements (black datapoints) were collected from independent cell passages, the mean of which is denoted by bar height. Green arrows indicates growth that was lower than expected, based solely on the number of DMOG treatment days. Significance ($*P < 0.05$, $**P < 0.01$) was determined by one-sided $t$ test for $\log_2$-transformed growth. This evaluated whether growth after the second protocol was below the value expected from an interpolation of the remaining three protocols to the number of days in DMOG, without considering treatment order. Source data are available for this figure: SourceData F5.

block (Exp5), where normoxic incubation was at pHe 6.4. Thus, an early period of strong HIF-1 α activation, followed immediately by normoxic acidosis, primed for poor growth (green arrows; Fig. 5 C). This is explained by a DMOG-orchestrated metabolic shift from respiration to fermentation, which leaves cells vulnerable to a subsequent period of acidosis blocking glycolysis while respiration is yet to recover from HIF-dependent downregulation. This growth-decelerating outcome is avoided when HIF-1α stabilization is attenuated under acidosis.

## HIF-1α is stabilized transiently under acidotic hypoxia

The effect of pHe on HIF-1α stabilization may emerge from the onset of treatment, or develop gradually. These alternatives were investigated by probing HIF-1α levels at 4, 16, or 48 h of hypoxic treatment at pHe 6.4 or 7.4 in SW1222 cells. Under alkaline conditions, HIF-1α stabilization was not significantly different between 4 and 48 h of treatment (Fig. 6 A). In contrast, acidotic hypoxia evoked a transient HIF-1α response that decreased significantly between 16 and 48 h of treatment (Fig. 6 B). HIF-1α levels still attained a prominent early peak if acidosis was applied 24 h prior to hypoxia, i.e., a preemptive acidosis (Fig. 6 C). This finding indicates that acidosis does not prevent HIF-1α protein from being produced. Overall, the data suggest that a degradative process is activated when acidosis and hypoxia are combined. The last time point with no significant difference in HIF-1α between acidotic and alkalotic hypoxia was determined empirically to be 16 h (Fig. 6 D), indicating that the HIF-1α degrading process is engaged with a delay. Consequently, certain early and stable transcriptional responses to HIF-1α may persist irrespective of pHe, whereas those with a slower onset are more likely to manifest pHe dependence. To rule out decreased *HIF1A* transcription as a cause for the pHe sensitivity, the effect of acidosis on *HIF1A* expression was studied by RT-qPCR in C99 and SW1222 cells. Relative to alkalotic hypoxia, *HIF1A* transcript levels were not decreased by acidotic hypoxia (Fig. 6 E). Taken together, our data suggest a posttranslational mechanism by which acidity triggers HIF-1α degradation, such as the canonical proteasomal (Jaakkola et al., 2001) or non-canonical lysosomal (Fujita et al., 2012; Hubbi et al., 2014; Hubbi et al., 2013; Lim et al., 2006; Liu et al., 2014) pathways.

## HIF-1α degradation under acidotic hypoxia is lysosome-dependent

As a first step toward distinguishing a proteasomal or lysosomal mechanism of HIF-1α degradation, we performed enrichment analysis (EnrichR) for "Cell Compartment" ontologies among differentially abundant proteins under acidotic hypoxia, reasoning that this may identify selectively activated pathways. Among significant proteins (Fig. 7 A and Table S5), those upregulated selectively under acidotic hypoxia associated with the "autolysosome" cell compartment, but not with the proteasome complex. The combined effect of acidosis and hypoxia on lysosomal-associated proteins is visualized on the heatmap for significant proteins featuring a lysosomal annotation (Fig. 7 B and Table S2). Our analysis revealed higher abundance of lysosome-associated proteins under acidotic hypoxia (Fig. 7 C). To verify that lysosomal activity is responsible for augmented HIF-1α degradation at low pH, HIF-1α stabilization was measured in the presence of the V-type ATPase inhibitor bafilomycin-A1, which uncouples the functionally significant luminal acidosis (Fujita et al., 2012; Lim et al., 2006). In C99 (Fig. 8 A and Fig. S3 A) and SW1222 cells (Fig. 8 B and Fig. S3 B), bafilomycin-A1 treatment modestly but noticeably increased normoxic HIF-1α levels under alkaline conditions, indicating a role for lysosomes in setting the balance between constitutive HIF-1α production and degradation. Strikingly, bafilomycin-A1 weakened the interaction between pHe and hypoxic HIF-1α levels,

such that hypoxic induction of HIF-1α was no longer strongly affected by pHe. Collectively, these findings implicate a role for lysosomes in contextualizing hypoxic responses to the acid-base environment of cells.

Evidence for lysosomal involvement in HIF degradation does not exclude a supplementary role for the canonical pathway involving ubiquitination by pVHL and targeting to the proteasome (Jaakkola et al., 2001; Maxwell et al., 1999). We tested this in RCC4 cells, a renal cell carcinoma cell line carrying an inactivating *VHL* mutation that causes tonic HIF-1α stabilization even under normoxia (Maxwell et al., 1999). 48 h treatment with acidity (under normoxia) was still able to reduce HIF-1α expression in RCC4 cells, despite the lack of pVHL-triggered degradation, discounting this component from our mechanism (Fig. 8 C). HT29 cells treated with the pVHL inhibitor VH-298 (Qiu et al., 2018) produced a modest increase in HIF-1α levels under alkaline normoxia, yet hypoxic HIF-1α stabilization remained attenuated under acidic conditions (Fig. S3 C). Moreover, pVHL abundance in SW1222 cells was reduced by acidotic hypoxia, which argues against a role in acid-evoked HIF-1α degradation (Fig. 8 D).

The effect of acidosis on HIF-1α could, instead, involve processes downstream of pVHL. If this were the case, treatment with the proteasome inhibitor MG-132 is expected to overshoot HIF-1α levels under acidic conditions, yet this was not observed in C99 cells (Fig. 8 E). Furthermore, inhibition of proteasomal activity by either MG-132 or epoxomicin, as confirmed by increased ubiquitin (Fig. 8 F), did not perturb the inhibitory effect of low pHe on hypoxic HIF-1α stabilization in C99 cells (Fig. 8 G). To directly interrogate proteasomal activity, we performed a luciferase-based assay after treating C99, SW1222, HT29, or HDC111 cells over a range of pHe, with and without DMOG (Fig. 8 H). After 16 h of treatment, we observed no difference in activity readouts, arguing against proteasome involvement in acid-triggered destabilization of HIF-1α under hypoxia. Overall, our evidence favors a lysosomal mechanism of HIF-1α degradation under acidotic hypoxia.

## HIF-1α degradation under acidotic hypoxia relates to a pro-autophagic phenotype

The ensemble protein-degrading capacity of lysosomes is the product of their organelle-level activity (inhibited by bafilomycin-A1) and abundance (Yim and Mizushima, 2020). The latter was studied by live-cell imaging using LysoBrite, an acidophilic fluorescent dye that accumulates in organelles of low luminal pH. For this purpose, C99 cells were selected because they grow as flat monolayers conducive for imaging. Incubations with various combinations of pHe and pO$_2$ were terminated at 4, 16, or 24 h prior to imaging. Absolute background signal, after Gaussian filtering, was used to demarcate cytoplasmic areas, whereas thresholded Hoechst fluorescence identified nuclei and enabled cell segmentation by waterfall methods. An exemplar set of images, taken at 24 h, is shown in Fig. 9 A (Fig. S4 A shows lower magnification and each time interval). Circular particle detection by a two-stage Hough transform was user-trained (using a randomly assigned subset of 10% images) to identify lysosomes based on brightness and diameter (Fig. S4 B). Images

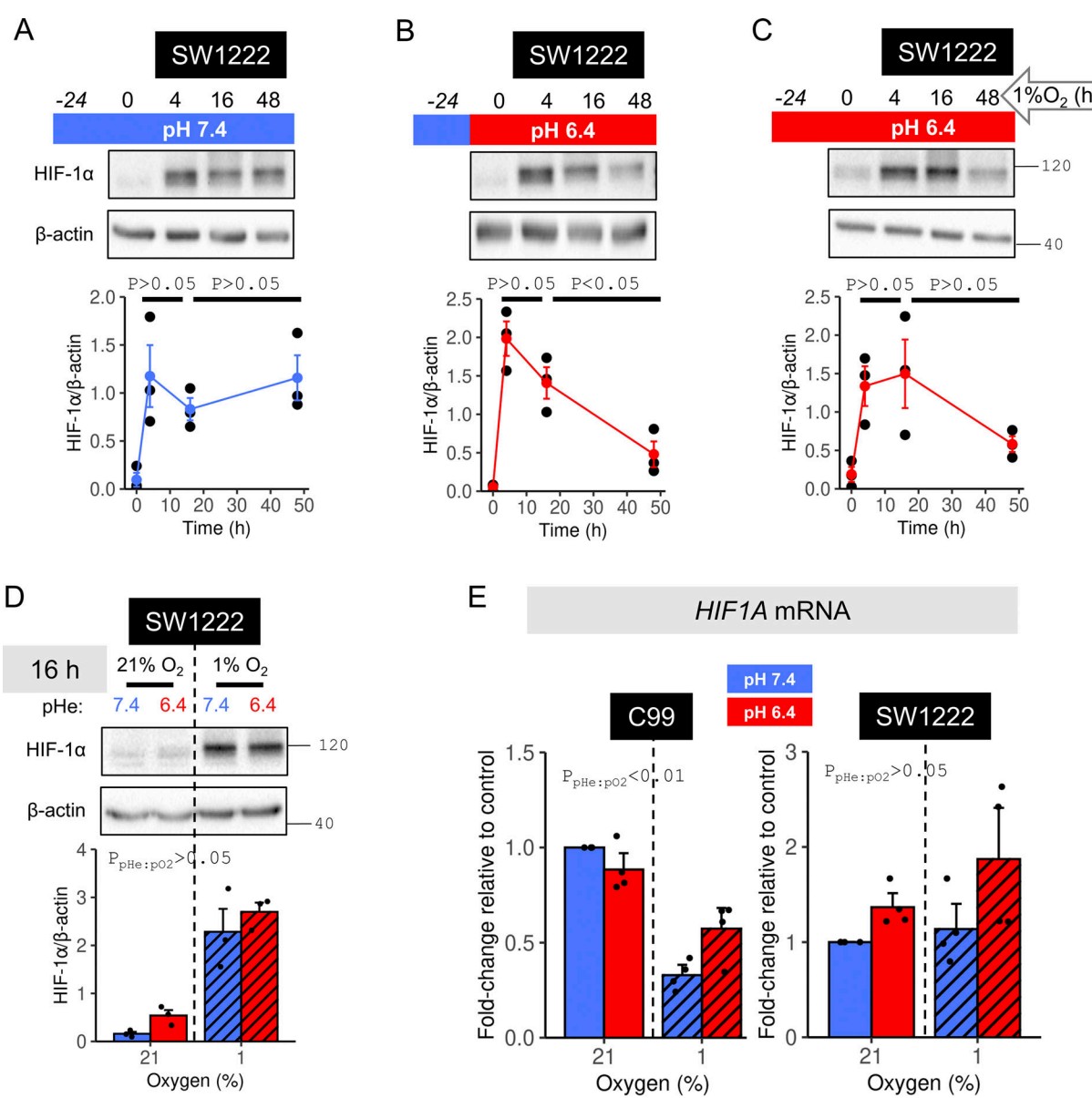

Figure 6. **Acidotic hypoxia evokes a time-dependent decay of HIF-1α protein. (A–C)** SW1222 cells were incubated for 24 h under normoxia (21% O₂) followed by a period of up to 48 h under hypoxia (1% O₂). Media were set to either (A) alkaline (pH 7.4) throughout the protocol, (B) alkaline during normoxia but acidic (pH 6.4) during hypoxia, or (C) acidic throughout the protocol. **(D)** SW1222 cells were exposed to 16 h normoxia or hypoxia in media at pH 6.4 or 7.4. **(A–D)** After treatments, lysates were analyzed for HIF-1α immunoreactivity. HIF-1α signals were normalized to loading control (β-actin) for three independent repeats. **(E)** RT-qPCR for *HIF1A* mRNA in C99 or SW1222 cells exposed to 48 h normoxia or hypoxia at either pHe 6.4 or 7.4. Fold-change relative to alkalotic normoxia calculated using the ΔΔC_T method with *ACTB* as the housekeeping gene (four independent repeats). Datapoints indicate individual repeats, and bars indicate mean ± SEM. Statistical testing by (A–C) one-way ANOVA with Tukey's test for multiple comparisons or (D and E) two-way ANOVA (see Table S1 for full results). Source data are available for this figure: SourceData F6.

were quantified in terms of the number of lysosomes per cell. At all time points, hypoxia significantly increased lysosomal number (Fig. 9 B), whereas acidosis did not significantly affect abundance as an independent factor, except at 4 h. A significant interaction between pHe and pO2 emerged by 24 h of treatment, such that acidotic normoxia reduced lysosomal abundance, but the opposite response was observed with acidotic hypoxia.

The observed effects on lysosome abundance were not accompanied by a substantial change in their diameter, identified through particle analysis (Fig. 9 C). However, combining

acidosis with hypoxia for 24 h shifted lysosomes further away from the nucleus, seen as a right-shift of the frequency distribution of lysosome-nucleus distance (Fig. 9 D). Given that the number of nuclei identified per imaging field of view did substantially change under acidotic hypoxia (Fig. 8 E), the redistribution of lysosomes likely reflects movement toward the cell periphery. It was previously been reported that the position of lysosomes within the cytoplasm modulates luminal pH, with possible consequences for proteolytic activity (Steffan et al., 2009). To test this, activity of cathepsin B (CTSB) was

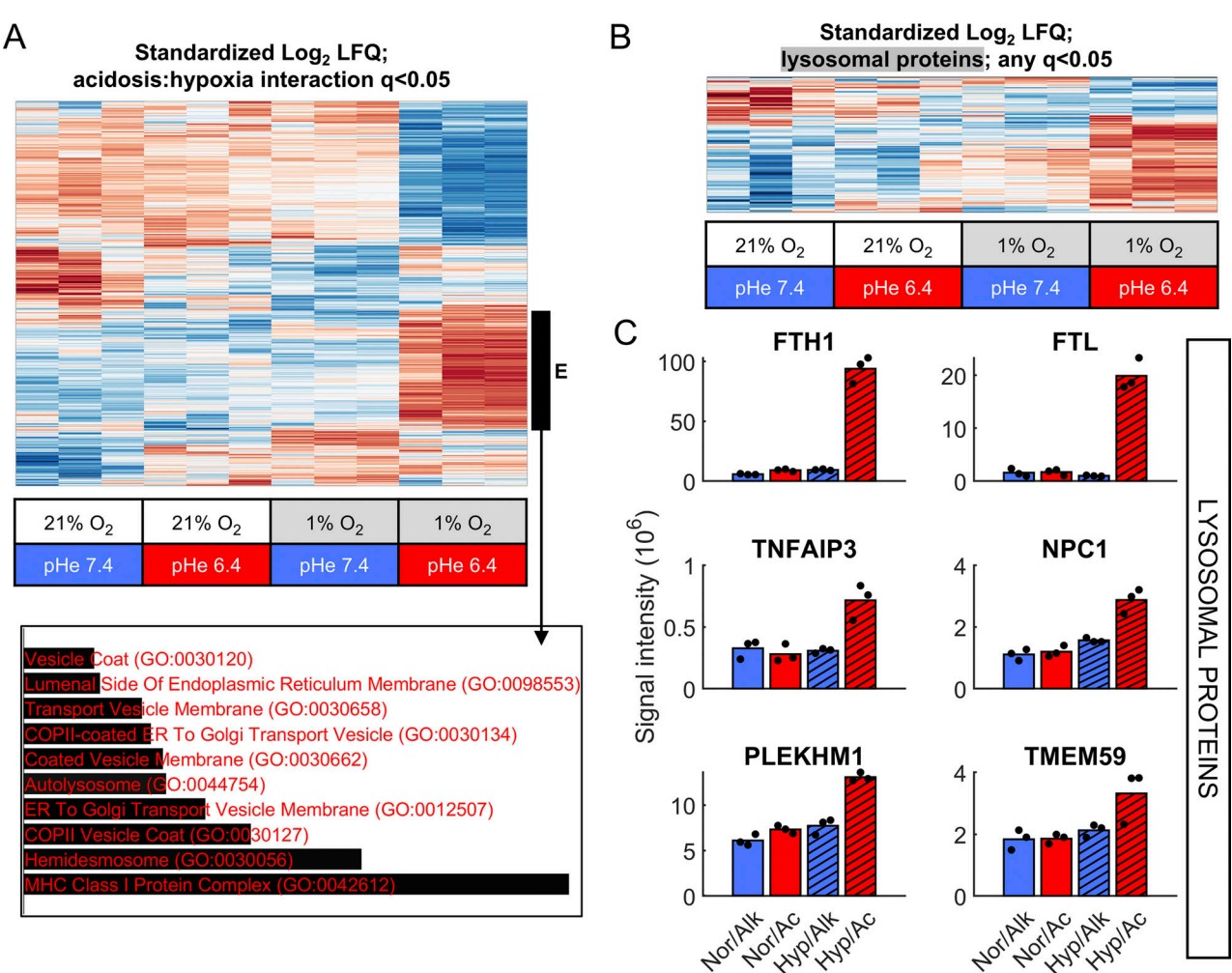

**Figure 7. Lysosomal pathway proteins are enriched under acidotic hypoxia. (A–C)** Proteomic analyses of SW1222 cell lysates collected after 48 h treatment at pH 6.4 or 7.4 under normoxia (21% $O_2$) or hypoxia (1% $O_2$) for three independent replicates. **(A)** Heatmap of $\log_2$-transformed standardized LFQ abundance for proteins with significant interaction between acidosis and hypoxia (q < 0.05). Clustering identified a group of proteins ("E") that increase in abundance selectively under acidic hypoxia. **(B)** Enrichment analysis (EnrichR) for cell compartment ontologies identifies autolysosome. Bar length indicates combined score of P value and odds ratio. Heatmap of differentially abundant proteins with a lysosomal GO annotation (q < 0.05 for effect of acidosis, hypoxia, or interaction). **(C)** Signal intensity of exemplar proteins associated with (auto)lysosomal processes; bars indicate mean, and black datapoints indicate individual replicates. Conditions labeled as Nor (normoxia), Hyp (hypoxia), Ac (pH 6.4), and Alk (pH 7.4). LFQ, label-free quantification.

directly measured using fluorescence emitted from the cleavage product of Magic Red substrate MR-(RR₂) (Hämälistö et al., 2020) under the different combinations of pHe and pO₂. Strikingly, pHe 6.4 and 1% O₂ synergized to increase the number of fluorescent particles per segmented cell (Fig. 9 F). Fluorescence was inhibited by bafilomycin-A1, confirming that the major source of captured CTSB activity was lysosomal. Consistently, bafilomycin-sensitive cleavage of pro-CTSB into CTSB was greater under acidotic hypoxia, relative to alkalotic hypoxia (Fig. 9 F, inset). Overall, our findings suggest that the combination of hypoxia and acidosis primes cells for increased degradative activity through lysosomes.

Lysosomes mediate the terminal step of autophagy, the process for recycling organelles and proteins (Yim and Mizushima, 2020). It is plausible that elevated autophagy is desirable under acidotic hypoxia because low pH and O₂ depletion block fermentation and respiration, respectively i.e., a collapse of two

major biosynthetic supply pipelines (Fig. 1 E). To test if acidotic hypoxia enhances autophagy, we measured the abundance of LC3-II, the form of microtubule-associated protein 1 light-chain 3 anchored to the membranes of autophagosomes: double-membrane organelles engulfing cellular components to which lysosomes fuse (Tanida et al., 2008). Strikingly, this marker showed a profound increase under acidotic hypoxia, relative to alkalotic hypoxia or acidotic normoxia (Fig. 10 A). If LC3-II localizes to the internal autophagosome membranes, degradation may occur upon lysosome fusion (Tanida et al., 2008) Thus, to ensure that LC3-II abundance reflects autophagosome formation without the confounding effects of lysosomal fusion, bafilomycin-A1 was used to inhibit autophagosome-lysosome fusion (Sharifi et al., 2015). As expected, bafilomycin-A1 increased LC3-II abundance across all tested combinations of pHe and pO₂ (Fig. 10 B). Critically, LC3-II levels were strongly elevated when acidosis and hypoxia were combined, relative to

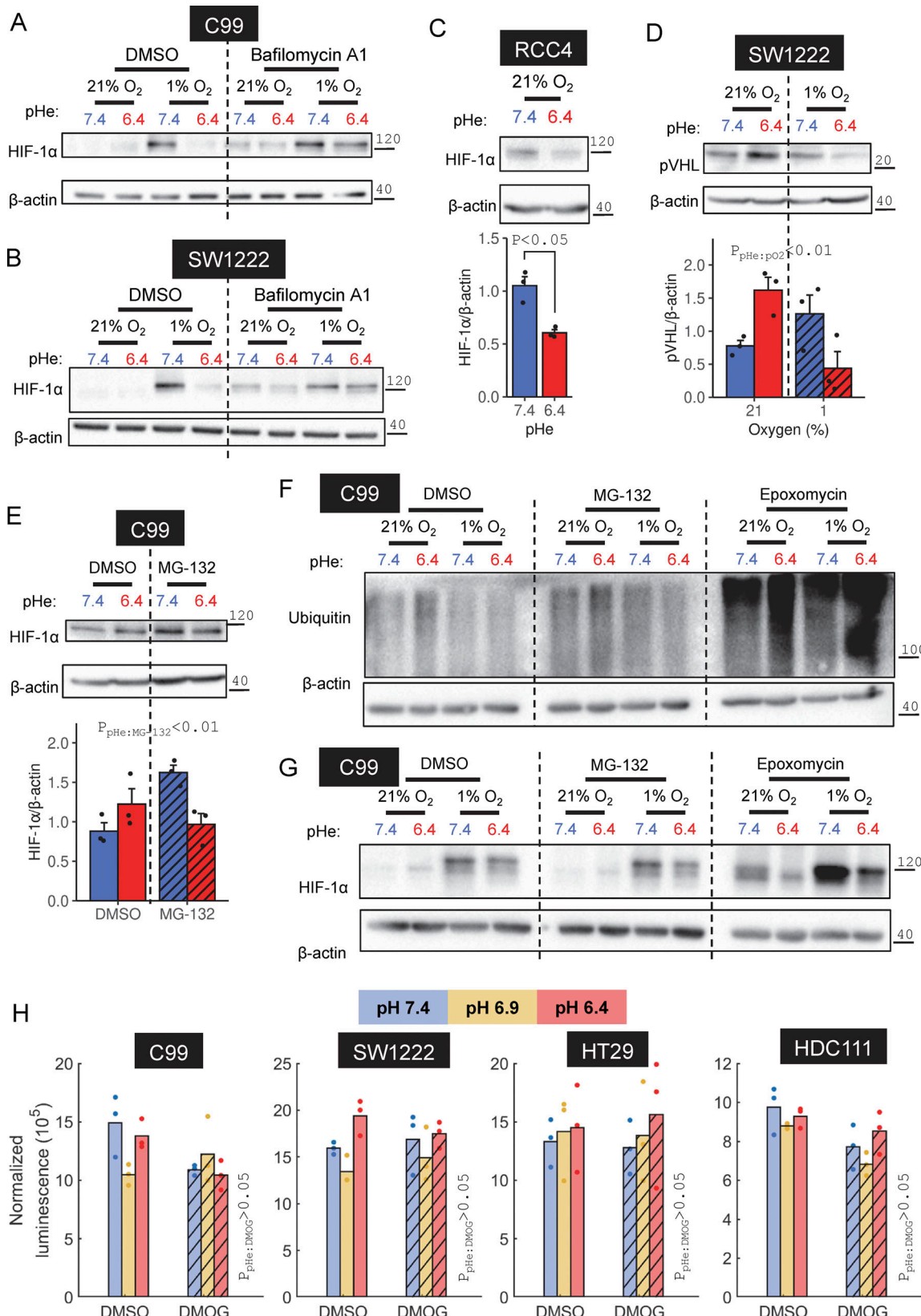

Figure 8. **HIF-1α degradation under acidotic hypoxia is lysosomal dependent. (A and B)** C99 and SW1222 cells were treated with normoxia (21% O₂) or hypoxia (1% O₂) in media at pHe 6.4 or 7.4 in the presence of DMSO or 20 nM bafilomycin A1. **(C)** VHL-null renal cell carcinoma cell line RCC4 was incubated in media at pH 6.4 or 7.4 under normoxic conditions. **(D)** SW1222 cells were treated with combinations of normoxia or hypoxia and pHe 6.4 or 7.4. **(E)** C99 cells were treated with vehicle control (DMSO) or 50 nM MG-132 in media at pH 6.4 or 7.4. **(F and G)** C99 cells were treated with combinations of normoxia or hypoxia and pHe 6.4 or 7.4, in the presence of DMSO, 50 nM MG-132, or 16 nM epoxomicin. After 48 h, lysates were analyzed for (A–C, E, and G) HIF-1α, (D)

pVHL, or (F) ubiquitin immunoreactivity. Where quantified, HIF-1α or pVHL signals were normalized to loading control (β-actin). **(H)** Proteasomal activity in C99, SW1222, HT29, and HDC111 cells was measured by luminescent assay following 16 h treatment at pHe 6.4, 6.9, or 7.4 with DMSO or 1 mM DMOG. Luminescence was normalized to the signal of cell-free control wells. Where quantified, experiments were performed for three independent replicates. Datapoints indicate individual repeats, and bars indicate mean + SEM. Statistical testing by (C) paired $t$ test or (D, E, and H) two-way ANOVA (see Table S1 for full results). Source data are available for this figure: SourceData F8.

when either factor was imposed separately. These results indicate that autophagy is enhanced when hypoxia is coupled with acidosis, but not with alkalosis.

Next, we used 3-methyladenine (3-MA) to probe whether autophagy relates to HIF-1α degradation under acidotic hypoxia. As a blocker of phosphatidylinositol 3-kinase, 3-MA disrupts autophagosome formation (Shi et al., 2020) and is predicted to rescue HIF-1α from degradation under acidotic hypoxia. However, multiple actions have been stipulated for 3-MA, thus its effect on hypoxic HIF-1α stabilization at low pHe must be contextualized to alkaline conditions, which corrects for pHe-independent background effects. Efficacy of 3-MA was confirmed by the increase in total LC3 abundance, suggestive of inhibited autophagosome-lysosome fusion (Sharifi et al., 2015) (Fig. S5 A). 3-MA reduced HIF-1α stabilization under alkalotic hypoxia but did not produce a proportional reduction under acidotic hypoxia (Fig. S5 B). In the presence of 3-MA, CA9 attained similar levels of hypoxic induction irrespective of pHe, indicating that a factor normally responsible for the pHe dependence of hypoxic signaling had become inactivated. The effect of 3-MA is unlikely to be mediated through mTOR signaling, as its reporters (S6K and S6 phosphorylation) were not substantially different to control conditions (Fig. S5 A). Combined with earlier evidence, we implicate a role for autophagosomal HIF-1α degradation selectively under acidotic hypoxia.

Next, we sought to identify the signaling pathways underpinning the selective enhancement of autophagic degradation under acidotic hypoxia. Expression of transcription factor EB (TFEB), a key regulator of lysosomal biogenesis (Settembre et al., 2011), increased under normoxic acidosis but not under hypoxic acidosis, i.e., the condition responsible for HIF-1α degradation (Fig. 10 C). Furthermore, TFEB knockdown did not rescue the inhibitory interaction between acidosis and hypoxia on HIF-1α, an observation that argues against TFEB signaling being essential for stimulating HIF-1α degradation under acidotic hypoxia. Since the mammalian target of rapamycin complex I (mTORC1) exercises a strong inhibitory influence over autophagy initiation (Jung et al., 2009), we assessed its status under acidotic hypoxia using Thr389-phosphorylated S6 kinase (pS6K) and Ser204/244 phosphorylation of ribosomal protein S6 (pS6) (Magnuson et al., 2012). In SW1222 (Fig. 10 D) and C99 (Fig. 10 E) cells, S6K and S6 phosphorylation were suppressed more strongly by acidotic hypoxia than by acidosis or hypoxia alone. To test an alternative readout of mTORC1 activation, we assessed Ser65 phosphorylation of eukaryotic translation initiation factor 4E-binding protein 1 (4E-BP1) (Woodcock et al., 2019) in SW1222 cells (Fig. S5 C). 4E-BP1 phosphorylation was attenuated by 48-h acidotic hypoxia, but not its components presented independently. These results show that acidotic hypoxia strongly disrupts mTORC1 signaling, a context permissive for autophagy.

The strong inhibition of mTORC1 under acidotic hypoxia leaves little space for further pharmacological inhibition (e.g., by rapamycin). If mTORC1 inhibition were a key trigger for HIF-1α degradation, then rapamycin should only produce a HIF-1α response when there is residual mTORC1 activity, i.e., under alkalotic hypoxia but not under acidotic hypoxia. To verify rapamycin efficacy, we showed similarly low levels of S6 phosphorylation across experimental groups (Fig. S5 A). Rapamycin moderately reduced HIF-1α abundance under alkalotic hypoxia, consistent with mTORC1 blockade, but did not have a proportional effect under acidotic hypoxia, ostensibly because its target is already inhibited by low pHe (Fig. S5 B). This evidence points to a role of mTORC1 inhibition in engaging autophagic degradation of HIF-1α.

## Discussion

Reduced oxygen availability is a powerful cue for cells to adapt to. Many of these responses involve HIF-operated transcriptional changes that suppress oxygen-dependent metabolism and play a well-established role in cancer biology because an oxygen-independent phenotype is considered advantageous within the hypoxic TME (Pugh and Ratcliffe, 2017). However, a complete appraisal of the benefits of hypoxic signaling must also consider the impact of the end-products of a remodeled metabolism. As hypoxic cells switch from respiration to fermentation, the chemical identity of excreted acid transitions from $CO_2$ to lactic acid (Firth et al., 1994; Kim et al., 2006). To compensate for the reduced ATP yield of fermentative metabolism, the HIF-mediated response is expected to acidify the TME substantially (Swietach et al., 2023). Consequently, the effect of engaging HIF signaling must consider its detrimental impact on TME pH, which may outweigh the benefits of an oxygen-independent phenotype. Our results show that this trade-off is regulated by the inhibitory effect of acidity on HIF-1α protein stability under hypoxia.

We find that acidotic hypoxia triggers an increase in lysosome abundance and disinhibition of an pro-autophagic phenotype by blocking mTORC1 (Jung et al., 2009). Since this process develops over several hours, the combination of acidosis and hypoxia leads to a HIF-1α transient, as the protein is first stabilized by hypoxia and then degraded by lysosomes primed for autophagy. Our evidence for a lysosomal mechanism is based on pharmacological disruption of lysosomal pH, found to stabilize HIF-1α, and a combination of imaging and protein-degradation assays that demonstrate enhanced lysosomal activity. Moreover, we firmly exclude a role for proteasomal mechanisms in contributing to the additional degradative activity unleashed by acidotic hypoxia. Previous studies have shown how stressful conditions reduce HIF-1α synthesis (Balukoff et al., 2020); such a mechanism would synergize with activated lysosomal degradation

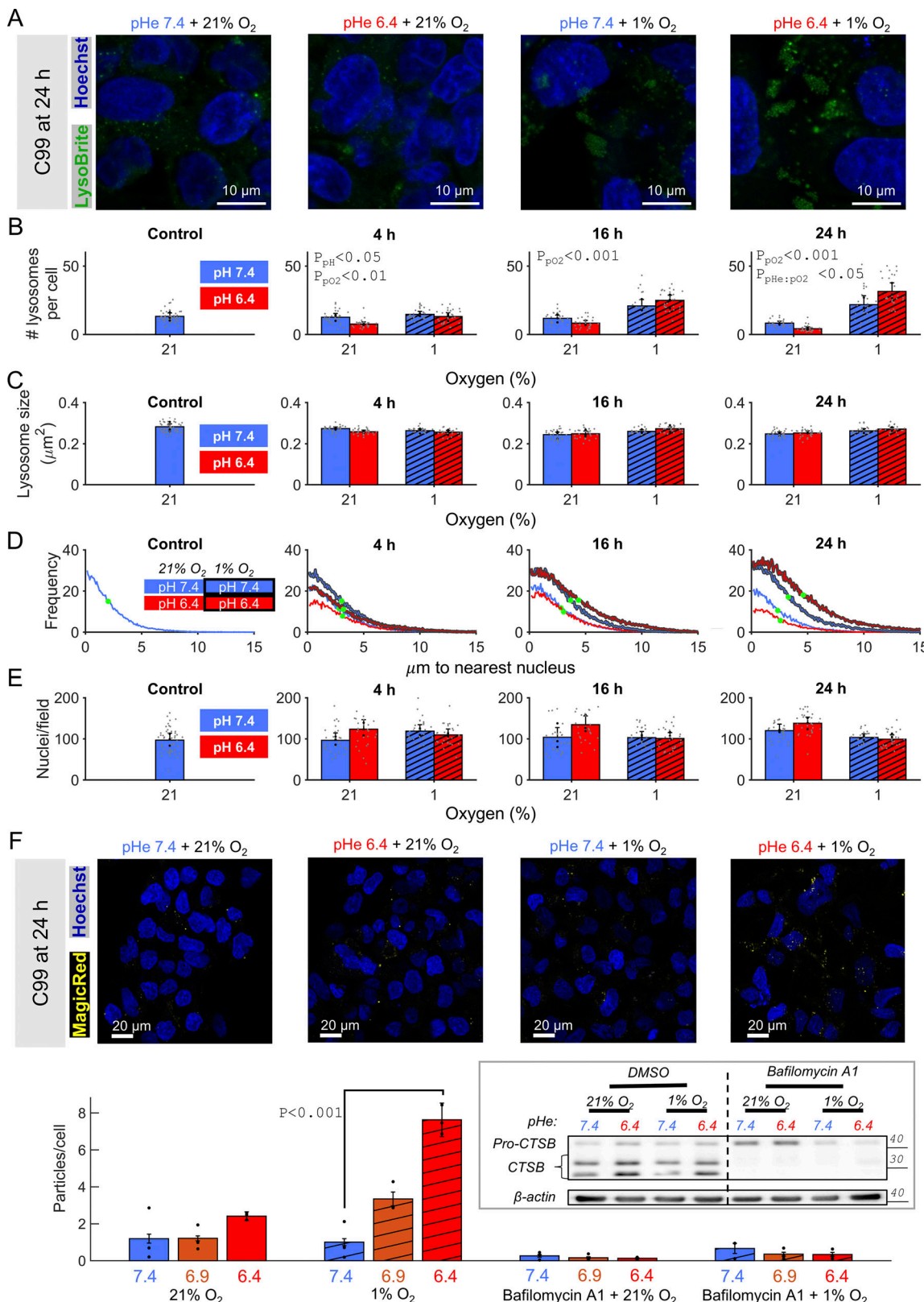

Figure 9. **Acidotic hypoxia increases lysosomal abundance.** CRC cells were incubated under normoxia (21% $O_2$) or hypoxia (1% $O_2$) in media at pH 6.4 or 7.4. **(A and B)** Treatment of C99 cells lasted up to 24 h and were followed by live-cell fluorescence imaging for Hoechst (nuclei) and LysoBrite Green (lysosomes) under normoxia and pHe 7.4. **(A)** Exemplar images at 24 h. **(B and C)** Quantification of LysoBrite–positive particles in terms of the number of lysosomes per cell (based on cell segmentation) and (C) lysosome size quantified as area. **(D)** Histogram of the distance from lysosome to its nearest nucleus, measured by applying a Euclidean distance transform to the binary image created from the segmented nuclear mask. Green dot indicates position at half-maximal abundance. **(E)** Quantification of the number of nuclei per field-of-view. **(B, C, and E)** Grey datapoints denote results from individual fields-of-view (three

independent repeats). Black datapoints indicate mean of each independent repeat. **(F)** C99 cells were treated with normoxia or hypoxia in media at pH 6.4 or 7.4 for 24 h. Magic Red-(RR)$_2$ was added at the start of incubations (1:260 dilution); Hoechst was added at the treatment end point, 30 min prior to imaging. Imaging sought evidence for fluorescence from the degradation product of Magic Red-(RR)$_2$. Quantification from three independent repeats, each representing the average of 10–20 images per condition. Inset: pro-CTSB cleavage after the 24 h treatment. 20 nM bafilomycin-A1 was added to inhibit lysosome activity. Bars indicate mean + SEM. Statistical testing by (B) hierarchical two- or (F) three-way ANOVA with multiple comparisons (see Table S1 for full results). Source data are available for this figure: SourceData F9.

in attenuating HIF-1α levels but would likely manifest as a reduced peak of HIF-1α, which we did not observe.

The combination of hypoxia and acidosis is highly relevant to tumors in vivo and is predicted to reduce the scope of HIF-1α–dependent responses. Our proteomic analyses, verified by immunoreactivity, showed that the expression of many canonical HIF targets, including CA9 and PDK1 (Choi et al., 2007; Kim et al., 2006; Rohani et al., 2019; Yeung et al., 2011), tracked HIF-1α levels and resulted in attenuated hypoxic responses under acidosis. However, the transient behavior of HIF-1α may result in different outcomes for rapid-onset and long-lasting transcriptional responses, potentially explaining why some hypoxic responses are not attenuated by acidosis. In terms of pathway-level outcomes, fermentative rate was upregulated by HIF stabilization under alkaline conditions but not under acidity, which can be interpreted as a safety mechanism to prevent acid overloading from an already acidic starting point. However, pathway-level hypoxic responses are not universally overridden by acidosis, as illustrated by respiration, where suppression persisted even at low pHe, likely because of reduced complex I assembly cancels the effect of PDK1 suppression. This compensatory response may be adaptive to prevent further O$_2$ depletion under already hypoxic conditions, irrespective of acid-base status. Overall, cells under acidotic hypoxia emerge with dually reduced fermentative and respiratory fluxes, causing a shortfall of biosynthetic materials and explaining why autophagy is favored specifically by the combination of hypoxia and acidosis (Yim and Mizushima, 2020). We argue that the resulting suppression of further acid production and oxygen depletion stabilize TME chemistry, during which cancer cells rely on recycling materials and metabolites.

Many of the responses, such as the emergence of LC3-II or suppression of mTORC1 markers, became strongly apparent with the combination of hypoxia and acidosis. Moreover, certain responses to hypoxia or acidosis were opposite to the effect of their combination (e.g., NDUFS1). These findings emphasizes the need to carefully consider whether an experiment using hypoxia also impacts pHe, and vice versa. Most experimental protocols have tended to introduce hypoxia before acidosis accumulates from the remodeled metabolism. This invariably produces a window for alkalotic hypoxia to trigger responses before the feedback from low pHe becomes meaningful. While this approach is valid for studying hypoxic signaling cascades experimentally, it is unlikely to be representative of tumors in vivo, where acidosis and hypoxia are linked by the common denominator of poor blood perfusion (Vaupel et al., 1989). In solid tumors, hypoxia is maintained by a diffusion barrier, which emerges from poor perfusion; that same barrier also restricts washout of metabolic acids. Consequently, pO$_2$ and pHe fall in tandem, albeit in a ratio that depends on a myriad of factors, including buffering capacity, the metabolic profile of cells, and how TME chemistry feeds back on phenotypes (Rohani et al., 2019; Vaupel et al., 1989). One prediction borne from our study is that some areas of true hypoxia, as revealed by pimonidazole staining, may not necessarily evoke HIF-1α responses if concurrent acidification curtails hypoxic signaling (Swartz et al., 2022). In other words, HIF-1α–positive regions may be a subset of hypoxic regions. Hypoxic areas that do not engage HIF signaling could be revealed by markers of acidosis (e.g., pHLIP) (Rohani et al., 2019; Weerakkody et al., 2013). Moreover, if hypoxia in tumor sections is gauged in terms of HIF-dependent responses (e.g., CA9 expression), the results could erroneously exclude regions where HIF became destabilised by acidosis. The use of HIF targets as markers of hypoxia (Olive et al., 2001; Rohani et al., 2019) should be considered with caution.

The ability of cancer cells to survive under hypoxia and acidosis is likely to impact their aggressiveness in vivo because TME chemistry is a powerful selection pressure. These adaptive mechanisms are therefore important to understand mechanistically. Whereas suppression of the canonical HIF effect on fermentation by acidosis can be explained in terms of protecting the TME from excessive acidification, the scope for pH to affect other HIF-dependent processes is likely to vary on a case-by-case basis. This is because the rationale for inhibiting hypoxic responses under acidosis is not universal, as shown by respiration and various individual protein responses. Specifically in the context of CRC, it would be important to understand whether the effect of hypoxia on vascular growth, a HIF-driven process, is also subject to acid inhibition. Improved vascular perfusion would reduce acidosis and hypoxia in tandem, thus synergy between the two chemical variables could be viewed as beneficial. Indeed, our proteomic data suggest that the hypoxic induction of VEGF (Liu et al., 1995) is potentiated under acidosis, unlike other canonical HIF targets. Further studies on vascular growth are warranted to verify functional outcomes and seek mechanisms. This information would be of clinical relevance to antiangiogenic therapies, such as bevacizumab, which target vascular growth to curtail CRC progression (Hurwitz et al., 2004). As we move away from a static view of TME chemistry, to one featuring complex spatiotemporal dynamics, the overall effect of hypoxia on transcription is likely to be less predictable than anticipated from in vitro studies.

## Materials and methods
### Cell lines
The study used human colorectal cell lines (C99, SW1222, HDC111, HT29, VACO5, DLD1, COLO678, and COLO320DM)

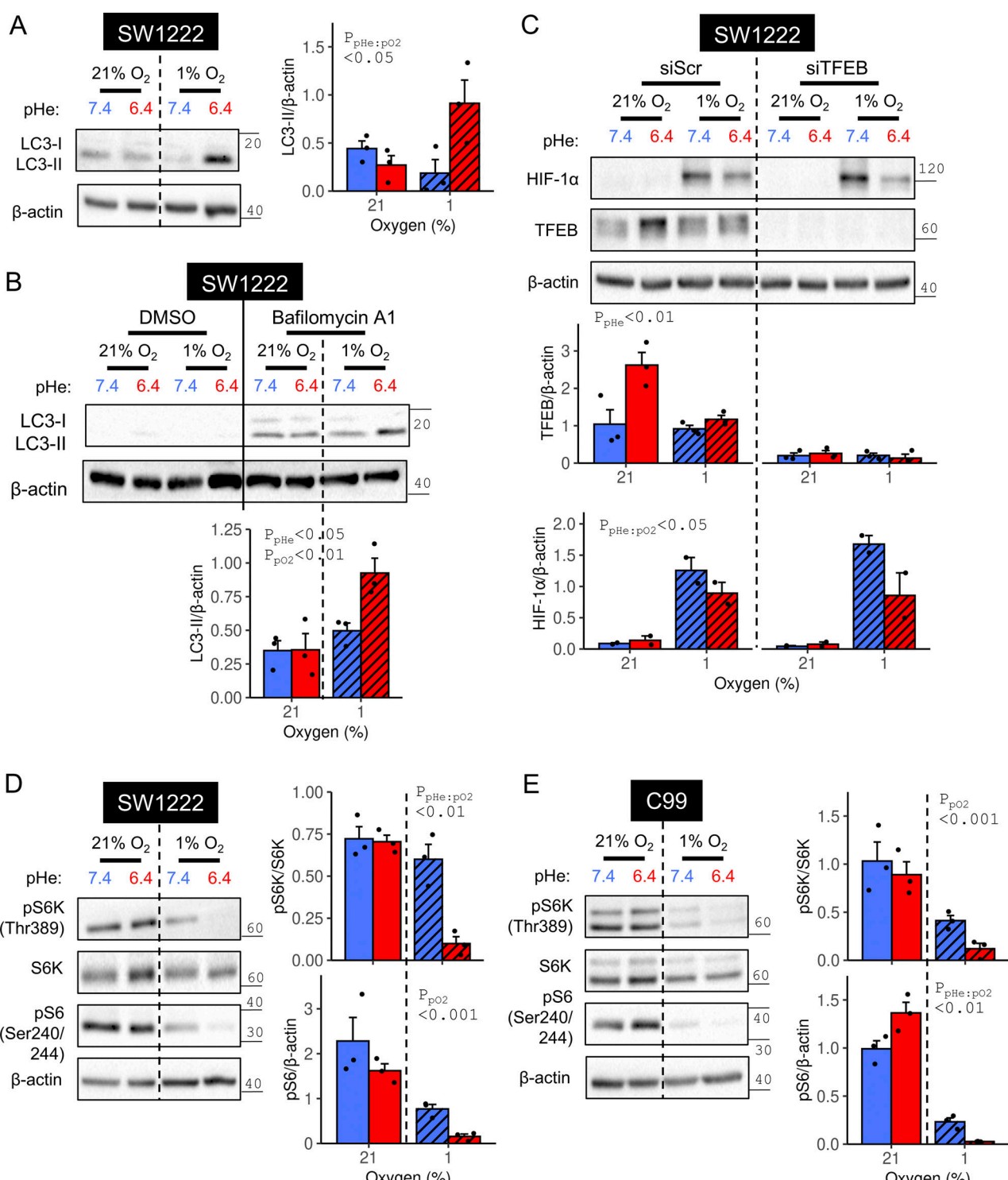

Figure 10. **Acidotic hypoxia promotes autophagy. (A)** SW1222 cells were incubated under (21% O$_2$) or hypoxia (1% O$_2$) at pHe 6.4 or 7.4. After 48 h of treatment, lysates were analyzed for markers of autophagy. LC3-II was normalized to loading control (β-actin). **(B)** Treatments were repeated in the absence or presence of 20 nM bafilomycin-A1 to suppress autophagosome-lysosome fusion. Quantification was performed for bafilomycin A1-exposed cells. **(C)** SW1222 cells were transfected with either non-targeting control siRNA (siScr) or siRNA-targeting *TFEB* (siTFEB). 24 h after transfection, cells were cultured for 48 h in media at pH 7.4 under either normoxia or hypoxia. Lysates were analyzed for TFEB and HIF-1α immunoreactivity, and signals were normalized to loading control (β-actin). **(D and E)** SW1222 or C99 cells were incubated under normoxia or hypoxia at pHe 6.4 or 7.4. After 48 h of treatment, lysates were analyzed for markers of mTORC1 signaling. pS6 signal was normalized to loading control (β-actin), and pS6K signal was normalized to S6K. Experiments were performed in three independent repeats. Datapoints indicate individual repeats, and bars indicate mean + SEM. Statistical testing by two- or three-way ANOVA (see Table S1 for full results). Source data are available for this figure: SourceData F10.

obtained from Professor Walter Bodmer's laboratory (University of Oxford) and a renal cell carcinoma cell line (RCC4) obtained from Professor Sir Peter Ratcliffe's laboratory (Ludwig Institute for Cancer Research, Oxford, UK). These cells divide outside the body, have been deidentified, and are not relevant material under the Human Tissues Act (UK). Cell lines were authenticated by short tandem repeat profiling provided by the European Collection of Authenticated Cell Cultures and routinely tested for mycoplasma contamination. Cell line stocks were maintained in DMEM (11965092; Gibco), supplemented with 10% FBS (A5256801; Gibco), 1% PS (100 U/ml penicillin, 100 μg/ml streptomycin; 15140122; Sigma-Aldrich), and 100 μg/ml Normocin (ant-nr-2; InvivoGen).

## Culture conditions for (pre-)treatments
Cells were (pre-)treated with media based on Phenol Red–containing, $NaHCO_3$-free DMEM (D7777; Sigma-Aldrich) supplemented with 10% FBS and 1% PS. Where Phenol Red–free DMEM (D5030; Sigma-Aldrich) was required, this was supplemented as per Phenol Red–containing media plus 1% GlutaMAX (35050038; Gibco) and 25 mM D-(+)-glucose. Added [NaCl] was adjusted to maintain constant osmolarity while adjusting $[HCO_3^-]$ from 0 to 44 mM and $pCO_2$ to set to attain a desired medium pH, according to established guidelines (Michl et al., 2019). Where media contained lactate, 20 mM NaCl was replaced by 20 mM sodium lactate, which matches the decrease in $NaHCO_3$ for acidotic media. All cells were cultured at 37°C, 21% $O_2$, and 5% $CO_2$ unless stated otherwise. 1% $O_2$ was imposed by replacement of $O_2$ by $N_2$ within a hypoxia incubator. DMOG (71210; Cayman Chemical), MG-132 (B1793; Cambridge Bioscience), VH-298 (SML1896; Sigma-Aldrich), bafilomycin A1 (B1793; Sigma-Aldrich), rapamycin (SM83-5; Cell Guidance Systems), 3-MA (M9281; Sigma-Aldrich), and epoxomicin (A2606; APExBio) were reconstituted in DMSO and then diluted in media to concentrations indicated in figure legends.

## Cell seeding densities
Recognizing that acidity (with or without hypoxia/DMOG) profoundly reduces cell growth, initial cell density at seeding was adjusted empirically to ensure comparable end point cell confluency, hence variables such as overall growth factor or metabolite depletion, unless stated otherwise. To that end, seeding densities for acidotic and/or hypoxic conditions were increased relative to alkalotic normoxia in the following ratios (order: alkalotic normoxia, acidotic normoxia, alkalotic hypoxia, and acidotic hypoxia): 1:2:1:2.7 for C99 immunoblots; 1:2:1:2.8 for SW1222 or HT29 immunoblots; 1:1:1:2 for HDC111 immunoblots; and 1:1.6:1:2.4 for SW1222 RT-qPCR. Seeding densities (cells/well) for alkalotic normoxia were 250,000 for C99; 300,000 for SW1222 or HT29; and 100,000 for HDC111.

## Acid sensitivity of growth
4,000 cells/well were seeded in sterile, tissue culture–treated clear 96-well plates (3799; Corning). Cells were treated for 6 days in media set to pH between 6.2 and 7.7, after which sulforhodamine B (SRB) assay was performed. Using a custom MATLAB script (Michl et al., 2022), percentage growth at each medium

pH was calculated from SRB absorbance normalized to the maximum SRB absorbance interpolated from its pHe dependence. Data from independent biological repeats were fitted to a biphasic Hill-type curve, with a parameter describing maximum growth and four parameters describing independently cooperative activatory and inhibitory interactions (i.e., a binding constant and Hill coefficient for each interaction).

## SRB assay
Cells were fixed with trichloroacetic acid (60 min, 4°C). Fixed cells were washed with $H_2O$ four times, then stained with 0.057% SRB (230162; Sigma-Aldrich) in 1% acetic acid (30 min, room temperature). Residual SRB was removed by washing with 1% acetic acid four times, then 10 mM Tris base was added to dissolve SRB. SRB absorbance was measured at 520 nm using a Cytation 5 microplate reader (CYT5MV; Agilent).

## Metabolic profiling
Cells were cultured in sterile, tissue culture–treated 96-well black-wall/clear-bottom plates (1210013; Agilent) under pre-treatment conditions specified in figure legends. For control pre-treatment (pHe 7.4, no DMOG), cells were seeded at 35,000/well or 50,000/well. Seeding densities for other pre-treatments were empirically adjusted to provide equivalent live cell densities (across all conditions of a given cell line) at the end of the pre-treatment period. After 48-h pre-treatment, cells were loaded (12 min, room temperature) with 12.5 μM CTO (C34551; Invitrogen) to report live cell density. Residual CTO was removed by washing, and then metabolic profiling was performed using an established protocol (Michl et al., 2022). Briefly, cells were placed in media of low buffering capacity containing the pH- and $O_2$-sensitive fluorophores HPTS and RuBPY and covered by 150 μl mineral oil to impose a diffusion barrier that slows ingress of atmospheric oxygen. As cells fermented and respired, the resulting fall in medium pH and $pO_2$ was recorded over up to 17 h alongside CTO signal in a Cytation 5 microplate reader at 37°C with a dual gas controller set to 21% $O_2$ and 0% $CO_2$. Where indicated, changes in medium pH and $pO_2$ were converted to readouts of cumulative acid production and $O_2$ consumption.

## Lactate measurements
Media samples (100 μl per condition) were collected from underneath the oil barrier after 8 or 17 h of metabolic profiling. Lactate concentration was measured using a Pentra C400 Clinical Chemistry Analyzer (Horiba) with ABX Pentra Lactic Acid reagent (A11A01721; Horbida) and calibrated against standards following the manufacturer's instructions.

## Immunoblotting
Cells cultured in sterile, tissue culture–treated 6-well plates were washed with 1X PBS and then lysed by scraping in 4°C 1X radioimmunoprecipitation assay buffer (9806; Cell Signaling) containing inhibitors of proteases and phosphatases (4906837001 and 5892953001; Roche). Samples were centrifuged (20 min, 4°C, 17,850 RPM), and the pellet was discarded. Total protein concentration per sample was measured using a bicinchoninic acid assay. Lysates were mixed with Laemmli

buffer (1610747; Bio-Rad) and 2-mercaptoethanol to denature proteins. 10–30 μg total protein was loaded onto a 10% acrylamide gel. Electrophoresis was performed at 150 V in Tris/glycine/SDS running buffer. Afterward, transfer to methanol-activated polyvinylidene membrane was performed at 250–400 mA for 120 min in ice-cooled Tris/glycine transfer buffer. Membranes were blocked in 5% milk dissolved in 1X TBS with 0.1% Tween 20 (TBS-T), followed by primary antibody incubation. Residual primary antibody was removed by washing membranes in TBS-T Membranes were then incubated with secondary antibodies diluted 1:10,000 (vol/vol) in 5% milk (60 min). Secondary antibodies were goat anti-rabbit or anti-mouse HRP-conjugated IgG (G21234 and G21040; Invitrogen). TBS-T washes were repeated to remove residual secondary antibody. Membrane chemiluminescence was visualized on a ChemiDoc (Bio-Rad) after incubation with enhanced chemiluminescence substrate (Thermo Fisher Scientific). Where proteins of similar molecular weight were probed on the same membrane, membranes were incubated with stripping buffer (46430; Thermo Fisher Scientific) (15 min) before blocking and antibody reapplication. All steps performed at room temperature unless stated otherwise. Signals were quantified using Image Lab software.

### Primary antibodies

Primary antibodies were diluted in 3% bovine serum albumin in TBS with 0.02% $NaN_3$ and incubated overnight at 4°C. Host species, dilution (vol/vol), supplier, and commercial identifier as follows: β-actin (mouse, 1:2,000; HRP-60008; Proteintech), NDUFS1 (rabbit, 1:500; PA5–22309; Thermo Fisher Scientific), monoclonal antibody against the proteoglycan-like domain of CA9 (mouse, 1:500; kind gift from Professor Silvia Pastoreková, Biomedical Research Center of the Slovak Academy of Sciences, Bratislava, Slovakia), LDHA (rabbit, 1:500; 3582S; Cell Signaling Technology), HIF-1α (mouse, 1:500; 610958; BD Transduction Laboratories), monoclonal antibody against HIF-2α (mouse, 1:4; kind gift from Professor Sir Peter Ratcliffe's laboratory at the Ludwig Institute for Cancer Research, University of Oxford), pVHL (rabbit, 1:500; 68547S; Cell Signaling Technology), pT389-S6K (rabbit, 1:1,000; 9205; Cell Signaling Technology), S6K (rabbit, 1:1,000; 2708; Cell Signaling Technology), pS240/S244-S6 (rabbit, 1:1,000; 5364; Cell Signaling Technology), LC3 (rabbit, 1:500; PA1–16931; Invitrogen), PDK1 (rabbit, 1:1,000; 3062; Cell Signaling Technology), CEACAM6 (mouse, 1:2,000; MA5–24164; Invitrogen), CTSB (rabbit, 1:1,000; 31718; Cell Signaling Technology), pS65-4E-BP1 (rabbit, 1:1,000; 9451T; Cell Signaling Technology), 4E-BP1 (rabbit, 1:1,000; 9644S; Cell Signaling Technology), ATP6v1A (rabbit, 1:1,000; 39517; Cell Signaling Technology), and CDX1 (rabbit, 1:1,000; ab126748; Abcam).

### Sample preparation and LC-MS/MS proteomics

A total of 600,000 cells were seeded into sterile tissue culture–treated 6-cm Petri dishes (21.5-cm$^2$ growth area) for the following conditions: 7.4 at 1% and 21% $O_2$ and 6.4 at 21% $O_2$. For the 6.4 at 1% $O_2$ condition, 700,000 cells were seeded per 21.5-cm$^2$ dish to ensure adequate protein yield for proteomic analysis.

After 48 h of incubation, cells were rinsed twice with ice-cold PBS and immediately scraped into pre-cooled RIPA buffer (9806S; Cell Signaling Technology) supplemented with 4% SDS, protease and phosphatase inhibitors (Halt, 78441; Thermo Fisher Scientific), and Pierce Universal nuclease (37.5 U/ml; 88701; Thermo Fisher Scientific). 30 μg protein aliquots were reduced (10 mM Tris(2-carboxyethyl)phosphine) and alkylated (50 mM iodoacetamide) for 30 min at ambient temperature. Samples were subsequently processed by S-trap micro (Protifi) protocol according to the manufacturer's instructions with 1.5 h 47°C tryptic (Sequencing Grade, Promega) digestion with 1:25 enzyme to substrate ratio. Samples were analyzed by LC-MS/MS using an EvosepOne LC system (Evosep Biosystems) connected to an Orbitrap Astral Mass spectrometer (Thermo Fisher Scientific) equipped with a high-field asymmetric ion mobility spectrometry (FAIMS) Pro Duo interfaced with an EASY-Spray source (Thermo Fisher Scientific). Tryptic peptides (~100 ng) were loaded onto the Evotips and analyzed using the Whisper 40 samples per day (31 min gradient) using the Aurora Elite column (15 cm × 75 mm ID, 1 mm C18; IonOptics) with integrated emitter and heated at 50°C using the Ion Optics heater. The FAIMS Pro Duo interface was operated in standard resolution mode, with carrier gas flow rate of 3.8 liters/min and a compensation voltage of –48. The Orbitrap Astral was operated in positive mode using the data independent acquisition (DIA) mode. MS1 scans were acquired in the Orbitrap at 240-K resolution over m/z range from 380 to 980. The MS1 normalized AGC was set at 500% with a maximum injection time of 3 ms and a RF lens of 40%. DIA MS2 scans were acquired in the Astral mass analyzer with nonoverlapping windows of 2 Th with precursor scan range from 380 to 980 m/z (299 MS2 scans per cycle) with a maximum injection time of 3 ms (maximum duty cycle of 0.6 s). Normalized AGC was set at 500% and RF lens at 40%. Isolated precursors were fragmented in the HCD cell using 25% normalized collision energy.

### Proteomic analyses

Raw data were searched in DIA-NN v1.9.2 against an in silico predicted library previously generated from the UniProt human proteome database, plus common contaminants. Cysteine carbamidomethylation was set as a fixed modification, with tryptic (Trypsin/P) peptides permitted with a maximum of one missed cleavage. Mass accuracy was set to automatic inference for MS1 and 20 ppm for MS2. Match-between-runs and RT-dependent cross-run normalization were enabled, with contaminant peptides excluded from quantification. Analysis was performed using the DEP package (v 4.4) in R. Missing values were imputed by the minProb method (q = 0.01). Pairwise statistical testing for all four contrasts was performed with FDR correction (Benjamini–Hochberg). Two-way ANOVA was performed with batch correction and FDR correction using Limma (v3.36.2) in R.

### siRNA transfection

siRNAs were dissolved in sterile 1X PBS to generate 10 μM stocks and stored at –20°C. siRNAs were obtained from Dharmcon: SiScr (D-001206-13-05), siTFEB (M-009798-02-0005), and

siHIF1A (M-004018-05-0005). For transfection, 3.6 µl siRNA was mixed with 300 µl Opti-MEM (31985047; Gibco) in a sterile tube. In a separate sterile tube, 6 µl Lipofectamine RNAiMAX Transfection Reagent (13778075; Invitrogen) was mixed with 300 µl Opti-MEM. Both tubes were combined, and the mixture was incubated (10 min, room temperature). Afterward, the mixture was added to 400,000 trypsinized cells resuspended in 3 ml standard culture medium in a sterile tissue culture–treated 6-well plate and incubated for 24 h before plating for experiments.

## RNA extraction and quality control
Cells cultured in sterile, tissue culture–treated 6-cm² dishes were trypsinized and resuspended in standard culture medium. RNA was extracted from the cell suspension using the Zymo Research Quick-RNA Miniprep kit (R1054) following the manufacturer's instructions. Eluted RNA was stored at –80°C. Total RNA concentration per sample was quantified using the Promega QuantiFluor RNA System (E3310) following the manufacturer's instructions for a multi-well plate protocol with 2 µl undiluted sample. RNA purity was confirmed by 260/280 ratio using a NanoDrop Lite.

## RT-qPCR
cDNA synthesis was performed using the Bio-Rad iScript Advanced cDNA Synthesis Kit (1725037) with 1 µg input RNA following the manufacturer's instructions. qPCR was performed in a MicroAmp Fast Optical 96-Well Reaction Plate, 0.1 ml (4346907; Applied Biosystems). Reaction volumes per well 2 µl cDNA diluted 1:10 (vol/vol) in nuclease-free $H_2O$, 2 µl nuclease-free water, 5 µl TaqMan Universal PCR Master Mix (4304437; Applied Biosystems), 0.5 µl housekeeping gene (ACTB) primer, and 0.5 µl gene of interest primer. Taqman primers obtained from Applied Biosystems: ACTB (Hs01060665_g1), CA9 (Hs00154208_m1), HIF1A (Hs00153153_m1), and CDX1 (Hs00156451_m1). For every reaction mix, negative controls produced by RT of nuclease-free $H_2O$ were run. Plates were sealed and then run on a StepOnePlus Real-Time PCR System (4376600; Applied Biosystems) in triplicate. Data were analyzed using the ΔΔCt method and presented as fold-change relative to control conditions (Livak and Schmittgen, 2001).

## Proteasomal activity measurement
Proteasomal activity was assessed using the luminescence-based Proteasome-Glo Chymotrypsin-Like Assay (G8621; Promega), following the manufacturer's instructions. Briefly, SW1222, C99, HDC111, and HT29 were seeded at 5,000 cells/well in sterile, tissue culture–treated, white-wall/clear-bottom plates (3610; Corning Costar) and allowed to settle for 8 h. Settled cells were treated for 16 h as per figure legends using 200 µl/well phenol red–containing media. The following day, media were replaced with 100 µl/well bicarbonate- and DMSO/DMOG-matched Phenol Red–free media, with or without 8 µM epoxomicin for an additional 2 h. Afterward, 100 µl of reconstituted Suc-LLVY-Glo substrate was added to each well. Luminescence was recorded every 10 min for 4 h using a Cytation 5 microplate reader set to 37°C in air. Background signal from epoxomicin-treated wells was subtracted from the peak luminescence of untreated

controls, and values were normalized to cell growth using an SRB assay performed on a parallel identically treated plate.

## Lysosome imaging
Cells were seeded at a density of 250,000 cells/well in sterile, tissue culture–treated 4-well µ-slides (80426; Ibidi). After settling, cells were treated with time lengths of various $pO_2$ and medium pH combinations as indicated in figure legends. Next, cells were loaded (15 min, 37°C) with 1X LysoBrite Green (22643; AAT Bioquest) and Hoechst-33342 in $HCO_3^-/CO_2$-buffered media of pH 7.4 under 5% $CO_2$ and 21% $O_2$. Loading medium was replaced by $HCO_3^-/CO_2$-buffered media of pH 7.4 containing 1X LysoBrite Green for imaging at 5% $CO_2$, 21% $O_2$, and 37°C. LysoBrite Green fluorescence was imaged at 488-nm excitation and emission >510 nm. Hoechst fluorescence was imaged at 405-nm excitation and <490 nm emission. Imaging was performed at room temperature on a Zeiss LSM 700 confocal microscope using a ×40, NA 1.4 oil-immersion objective using ZEN software.

## Lysosome image analysis
Image analysis was performed in MATLAB using Hough Transform to detect circular particles within a tolerance of circularity, within a radius range, and meeting criteria for sufficient fluorescent signals. The algorithm was trained to identify the radius and intensity criteria of circular LysoBrite particles that are visually confirmed to be lysosomes. Training involve 50 random sample areas per image, with sufficient images used to reach convergence. During subsequent analyses, 10 fields-of-view were analyzed per condition per biological repeat. Background fluorescence on the UV channel was used to demarcate cell-occupied regions. The Hoechst-derived nuclear mask was used to segment nuclei by waterfall algorithms and obtain a count of cells per field-of-view for normalizing purposes, e.g., number of lysosomes per cell. Cytoplasmic particles meeting the criteria for lysosomes were counted, assessed for size and distance from nearest nucleus.

## Imaging CTSB activity
CTSB activity was captured using Magic Red Cathepsin B Kit (ICT937; BioRad) following the manufacturer's instructions. Briefly, cells seeded in sterile, tissue culture–treated 8-well slides (80826; Ibidi) were placed in treatment media (acidosis/alkalosis, with/without hypoxia, with/without 20 nM bafilomycin-A1) with Magic Red substrate MR-(RR)2 added at 1:260 dilution. After 24 h of treatment, cells were loaded with Hoechst-33342 (1:1,000, 30 min). Next, media were replaced with HEPES-buffered normal Tyrode of pH matching the treatment, 1 mM DMOG for hypoxia-treated cells only, and 20 nM bafilomycin-A1 for previously bafilomycin-A1–treated wells. Imaging was performed on a Zeiss LSM 700 confocal microscope at room temperature in air. Magic Red substrate MR-$(RR)_2$ fluorescence was imaged at 555-nm excitation and emission >600 nm. Hoechst fluorescence was imaged at 405-nm excitation and <490-nm emission. Particles were identified based radius matching that of lysosomes. Particles of mean fluorescence lower than 15% of saturating signal were pruned.

## Statistics

Sample sizes refer to the number of independent repeats (at least three were obtained for quantification). Data distributions for parametric tests were assumed to be normal, but this was not formally tested. Statistical testing as indicated in figure legends.

## Data, materials, and software availability

All study data are included in the article and/or SI Appendix. All raw data, details of methods, and analytical tools and scripts are available upon reasonable request to the corresponding author. Previously published data (Michl et al., 2024) were reanalyzed in Fig. 1 A to justify the use of cell lines in this study. The mass spectrometry proteomics data have been deposited to the ProteomeXchange Consortium via the PRIDE (Perez-Riverol et al., 2025) partner repository with the dataset identifier PXD062474.

## Online supplemental material

Fig. S1 shows the cell density measured by CTO. Fig. S2 shows the interaction between acidosis and hypoxia on protein levels. Fig. S3 shows the testing lysosomal mechanism of HIF-1α degradation. Fig. S4 shows the exemplar LysoBrite images and training of particle detection algorithm. Fig. S5 shows the testing mechanisms of lysosomal activation. Table S1 shows the results of statistical tests for data presented in figures. Table S2 shows the proteins detected by proteomics, including abundance and significance testing. Table S3 shows the differentially abundant proteins with significant effect of acidosis, hypoxia, or their combination. Table S4 shows the list of proteins that have a correlating pattern of abundance, across the experimental conditions, with HIF-1α (significant, $P < 0.05$; Spearman's test). Table S5 shows the list of proteins belonging to clusters A–E indicated in heatmaps of Figs 3 and 7.

## Data availability

All study data are included in the article and/or SI Appendix. All raw data, details of methods, and analytical tools and scripts are available upon reasonable request to the corresponding author. Previously published data (Michl et al., 2024) were reanalyzed in Fig. 1 A to justify the use of cell lines in this study. The mass spectrometry proteomics data have been deposited to the ProteomeXchange Consortium via the PRIDE (Perez-Riverol et al., 2025) partner repository with the dataset identifier PXD062474.

## Acknowledgments

We thank Professor Sir Walter F Bodmer (Oxford) for access to CRC cell lines, including their genetic and phenotypic characterization. We thank Syeeda Nashitha Kabir and Samira Lakhal-Littleton for access to lactate measurements.

This work was supported by a Cancer Research UK PhD studentship (PHDSTU-Hist\100200) to B. White and Bowel Research UK grant fund SG-24003 and European Research Council (SURVIVE #723997) to P. Swietach. Open Access funding provided by University of Oxford.

Author contributions: B. White: conceptualization, formal analysis, funding acquisition, investigation, methodology, project administration, supervision, validation, visualization, and writing—original draft, review, and editing. Z. Wang: investigation. M. Dean: investigation and validation. J. Michl: conceptualization, methodology, and supervision. N. Nieora: investigation. S. Flannery: formal analysis and investigation. I. Vendrell: formal analysis, resources, visualization, and writing—review and editing. R. Fischer: formal analysis, investigation, methodology, resources, and writing—review and editing. A. Hulikova: data curation, investigation, methodology, project administration, validation, and writing—original draft, review, and editing. P. Swietach: conceptualization, formal analysis, funding acquisition, project administration, resources, software, supervision, visualization, and writing—original draft, review, and editing.

Disclosures: The authors declare no competing interests exist.

Submitted: 24 September 2024

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

# Supplemental material

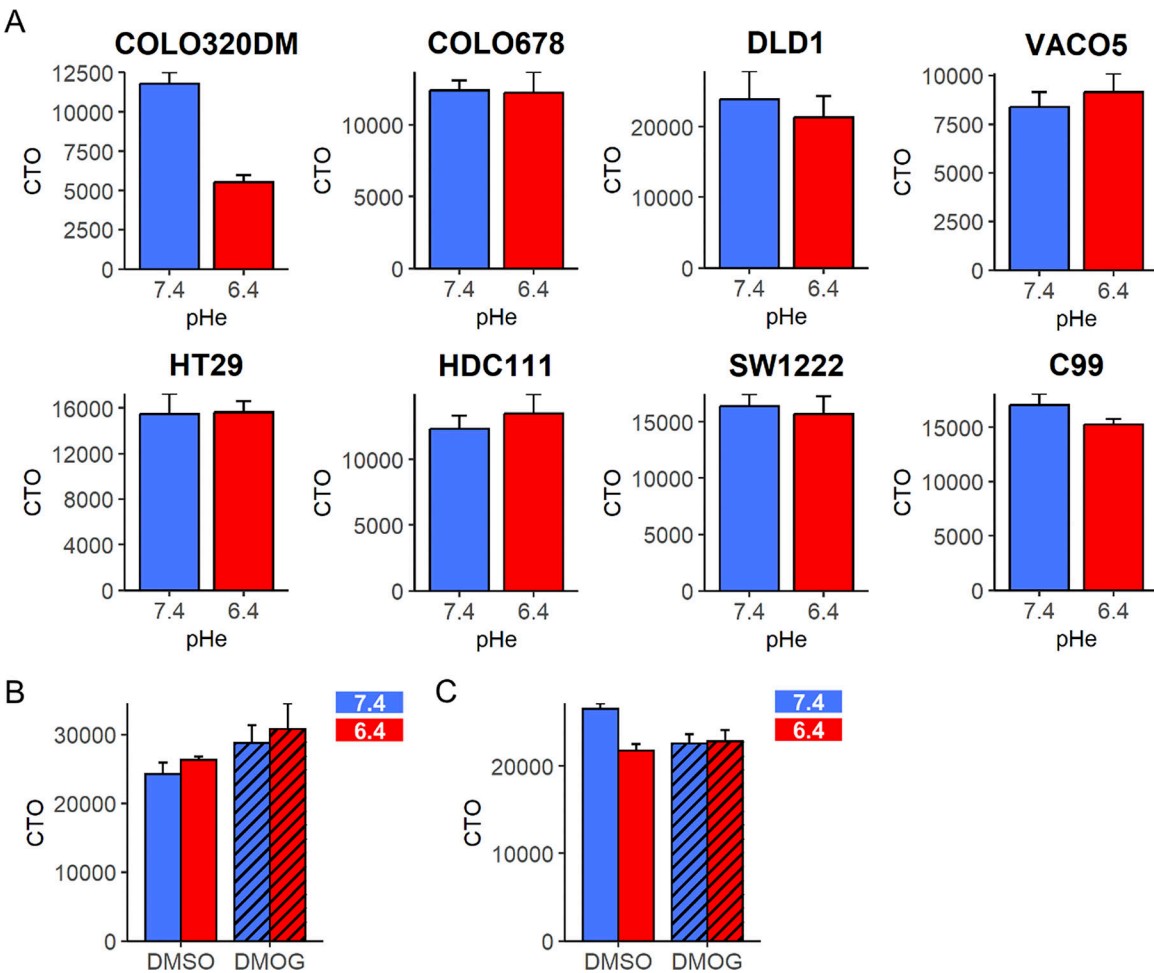

Figure S1. **Live cell density during metabolic profiling of CRC cell lines, inferred from CTO fluorescence retention (after loading and excess dye washout). (A)** Measurements in eight CRC cell lines that had been pre-treated for 48 h with either alkaline (pH 7.4) or acidic (pH 6.4) media. **(B and C)** Measurements in C99 cells following 48 h pre-treatment in alkaline or acidic media in the presence of 1 mM DMOG (hatched bars). Data presented as mean ± SEM.

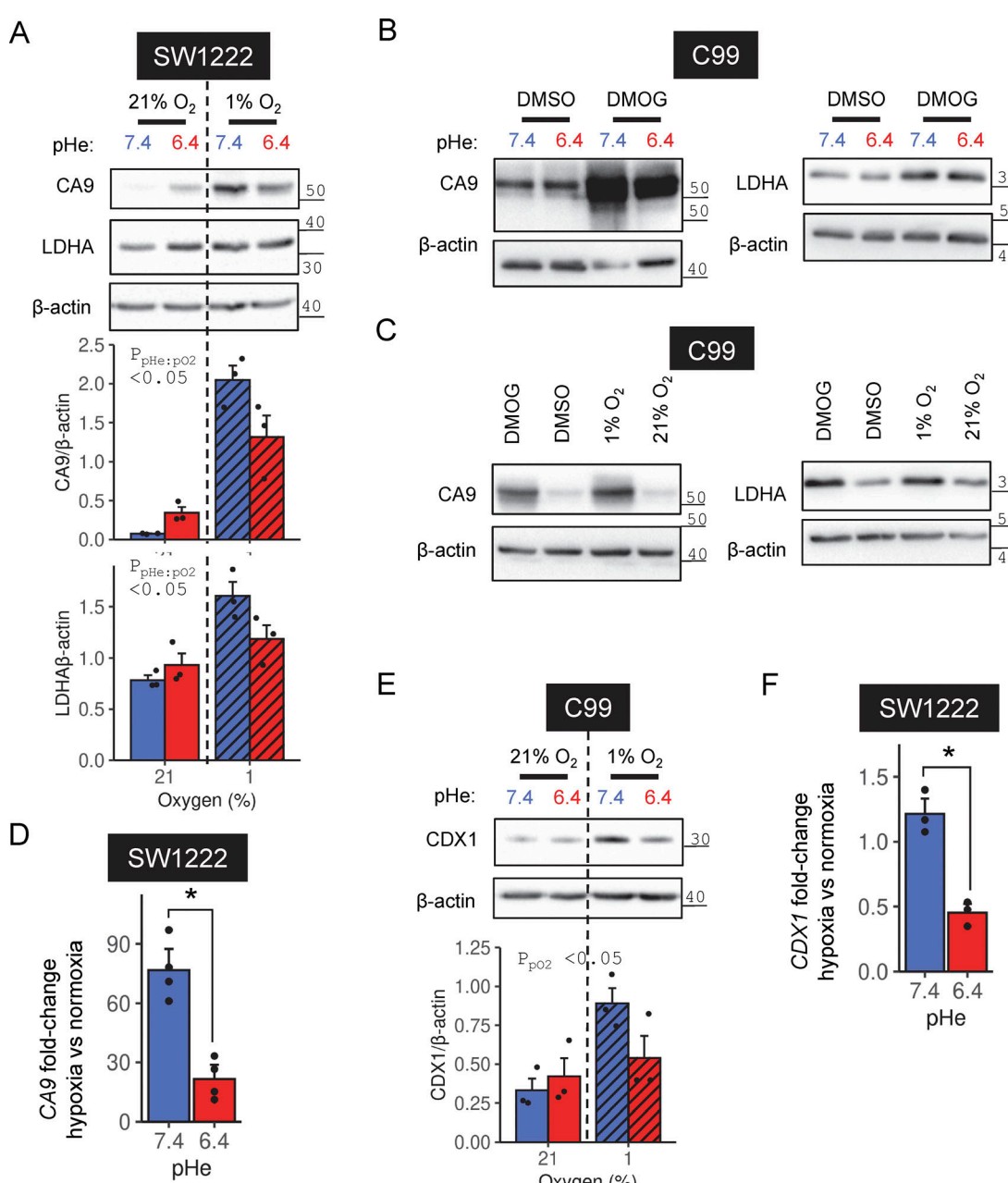

Figure S2. **pHe dependence of HIF-target induction. (A, B, and E)** SW1222 or C99 cells were grown in media at either pH 6.4 or 7.4 for 48 h. Concurrently, cells were exposed to either (A and E) normoxia (21% $O_2$) or hypoxia (1% $O_2$) or exposed to (B) DMSO or 1 mM DMOG. After treatment, lysates were collected and analyzed for immunoreactivity to (A and B) CA9, LDHA, or (E) CDX1. **(A and E)** Signals normalized to loading control (β-actin) for three independent repeats. Statistical testing by two-way ANOVA. **(C)** C99 cells were grown in media at pH 7.4 with either no additional treatment, DMSO treatment, 1% $O_2$, or 1 mM DMOG treatment for 48 h. Lysates were analyzed for CA9 and LDHA immunoreactivity. **(D and F)** SW1222 cells were grown at pH 7.4 or 6.4 under normoxia or hypoxia for 48 h, after which mRNA was extracted and RT-qPCR was performed for *CA9* or *CDX1* mRNA. Fold-change between hypoxia and normoxia calculated for each pHe treatment by ΔΔ$C_T$ method using *ACTB* as the housekeeping gene (three or four independent repeats). Statistical testing by paired *t* test. * indicates P < 0.05. Datapoints indicate individual repeats, and bars indicate mean ± SEM. See Table S1 for full results of statistical testing. Source data are available for this figure: SourceData FS2.

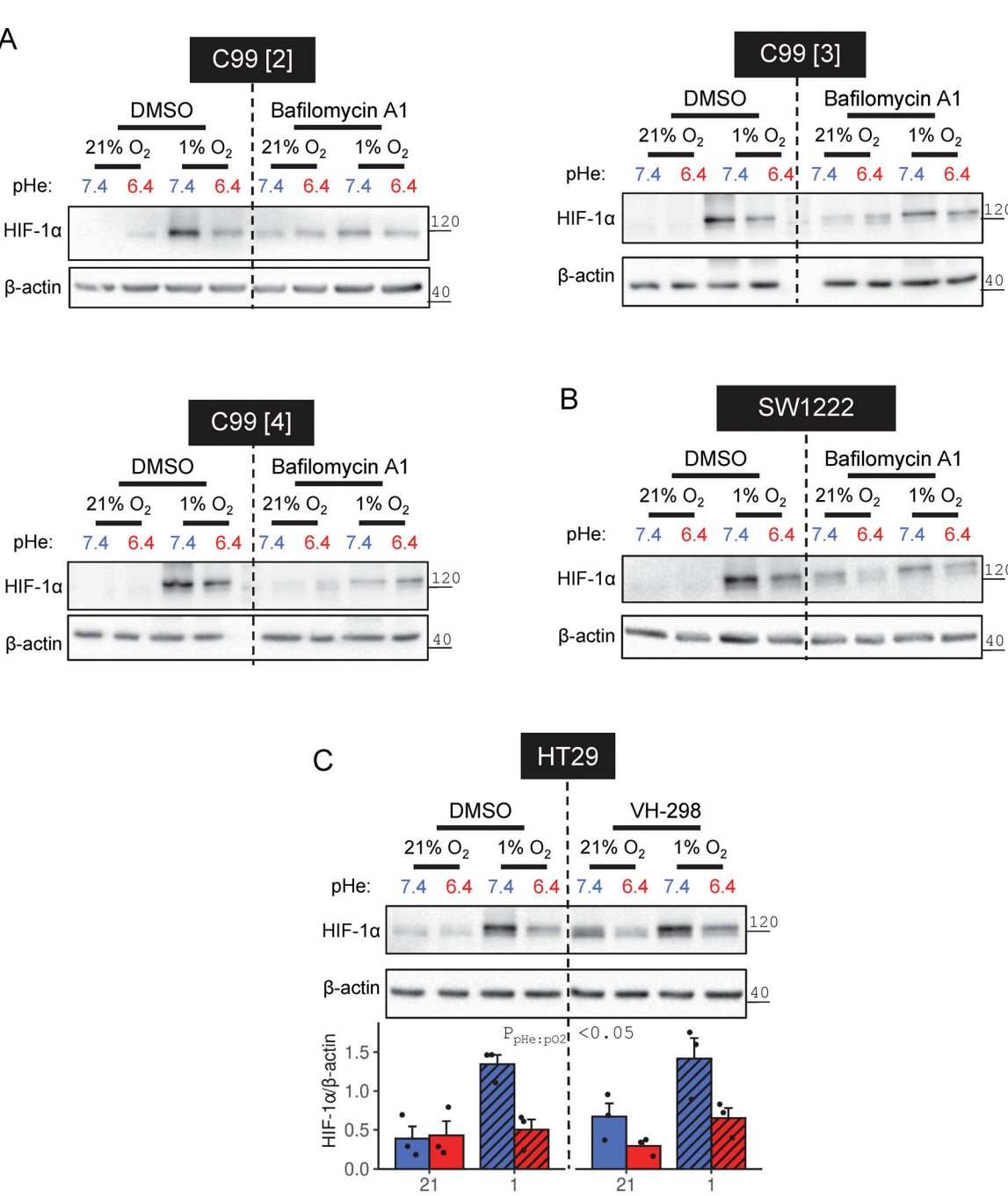

Figure S3. **Effects of proteasomal/pVHL and lysosomal inhibitors under acidotic hypoxia. (A–C)** SW1222, HT29, or C99 cells were incubated under normoxia (21% O₂) or hypoxia (1% O₂) in media at pHe 6.4 or 7.4, either in the presence of DMSO (vehicle control), (A and B) 20 nM bafilomycin-A1, or (C) 100 µM VH-298. After 48 h treatment, lysates were collected and analyzed for HIF-1α, ubiquitin, or β-actin immunoreactivity. **(A)** Three independent repeats displayed, in addition to the repeat shown in Fig. 8 A. **(C)** HIF-1α signals were normalized to loading controls (β-actin) for three independent repeats. Datapoints indicate individual repeats, and bars indicate mean ± SEM. Statistical testing by three-way ANOVA (see Table S1 for full results). Source data are available for this figure: SourceData FS3.

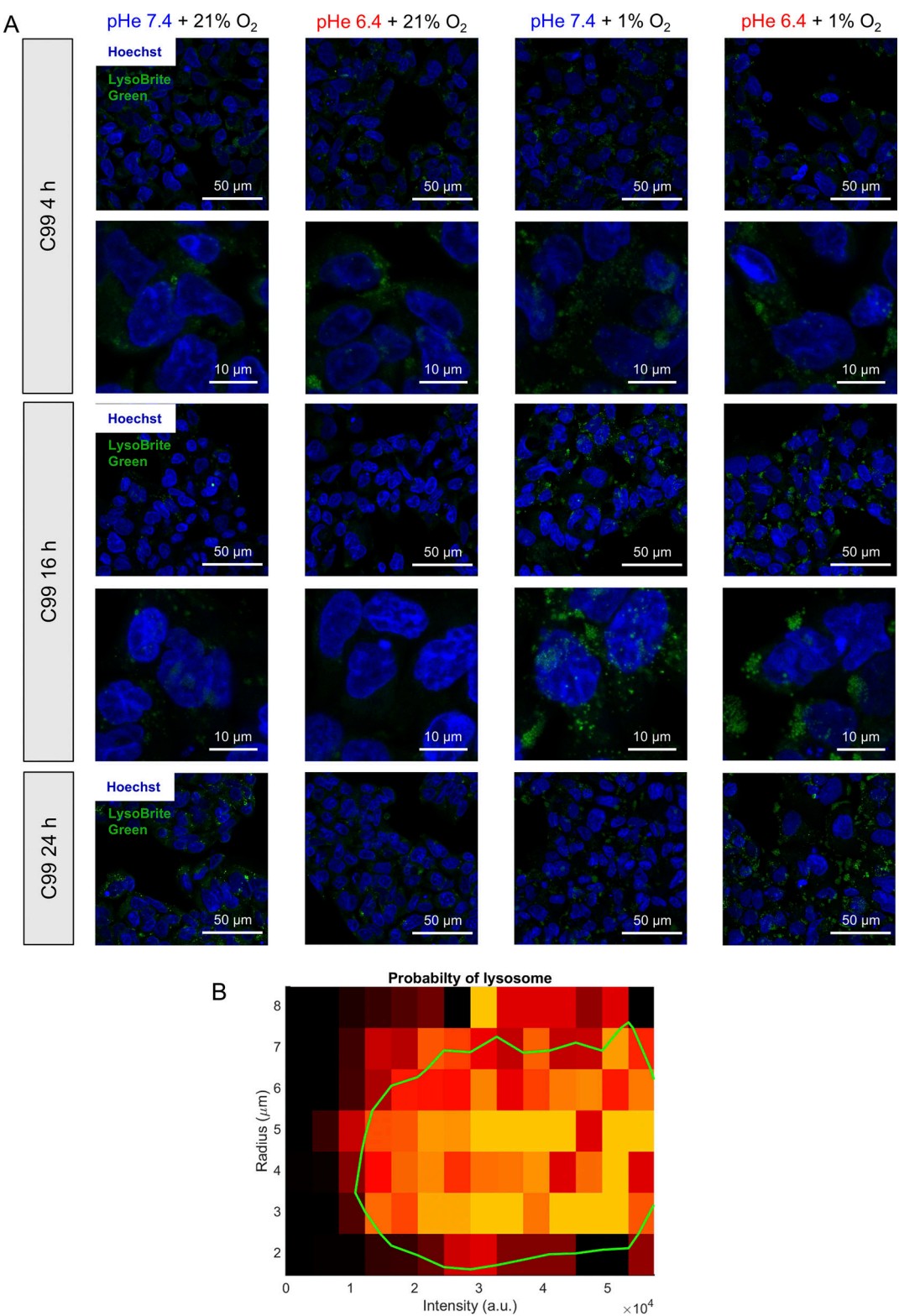

Figure S4. **Imaging lysosomes. (A)** Time course of lysosomal staining in response to combinations of acidosis and alkalosis with either hypoxia (1% $O_2$) or normoxia (21% $O_2$). Exemplar images of C99 cells, stained with Hoechst and LysoBrite Green prior to imaging. After treatment, live-cell fluorescence imaging was performed for nuclei (Hoechst) and lysosomes (LysoBrite Green) in imaging-compatible media at pHe 7.4 and normal atmosphere. Exemplar images are shown for low or high magnifications. Note, the high-magnification images at 24 h are shown in Fig. 8 A. **(B)** Criteria for classifying LysoBrite Green particles as lysosomes. Training used images selected at random and zoomed-in (>10 per image) for visual inspection. The software shortlisted particles that meet criteria of radius (range 2–10 pixels) and circularity of 1.0. Particles were presented to the inspector, who determined if the particle represents a lysosome or not. After repeating this process at least 6,000 times, data were summarized as a probability density map of radius and intensity to demarcate the 50% probability threshold (green line) defining criteria within which a particle is deemed to classify as a bona fide lysosome.

**White et al.**

Acidity attenuates hypoxic signaling

**Journal of Cell Biology** S4

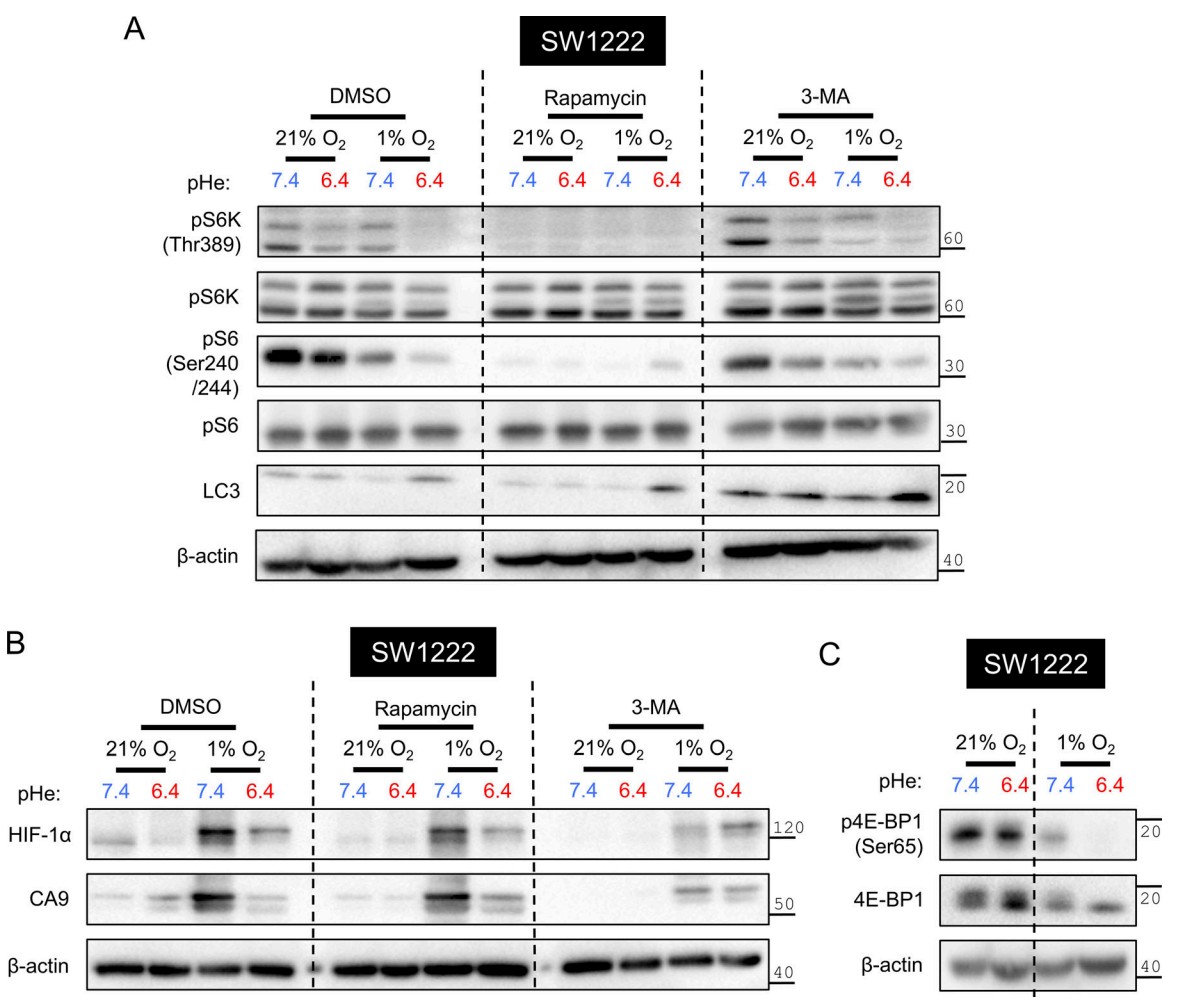

Figure S5. **Regulation of the HIF-1α pHe/pO$_2$ interplay by mTORC1 and autophagy.** SW1222 cells were incubated under normoxia (21% O$_2$) or hypoxia (1% O$_2$) at pHe 6.4 or 7.4 for 48 h. **(A–C)** Incubations in the presence of either DMSO, 100 nM rapamycin, or 5 nM 3-MA. After treatment, lysates were analyzed for markers of (B) HIF signaling or (A and C) mTORC1 signaling and autophagy. β-actin was used as a loading control. Source data are available for this figure: SourceData FS5.

**Provided online are Table S1, Table S2, Table S3, Table S4, and Table S5. Table S1 shows the results of statistical tests for data presented in figures. Table S2 shows the proteins detected by proteomics, including abundance and significance testing. Table S3 shows the differentially abundant proteins with significant effect of acidosis, hypoxia, or their combination. Table S4 shows the list of proteins that have a correlating pattern of abundance, across the experimental conditions, with HIF-1α (significant, P < 0.05; Spearman's test). Table S5 shows the list of proteins belonging to clusters A–E indicated in heatmaps of Figs 3 and 7.**

