## [Peer Review File · The Journal of Cell Biology]

Acidosis attenuates the hypoxic stabilization of HIF-1 α by activating lysosomal degradation

Bobby White, Zhenyi Wang, Matthew Dean, Johanna Michl, Natalia Nieora, Sarah Flannery, Iolanda Vendrell, Roman Fischer, Alzbeta Hulikova, and Pawel Swietach

Corresponding Author(s): Pawel Swietach, University of Oxford

Review Timeline:

Submission Date:	2024-09-24
Editorial Decision:	2024-10-25
Revision Received:	2025-04-12
Editorial Decision:	2025-05-13
Revision Received:	2025-05-19

Monitoring Editor: Johan Auwerx

Scientific Editor: Andrea Marat

Transaction Report:

DOI: <https://doi.org/10.1083/jcb.202409103>

October 25, 2024

Re: JCB manuscript #202409103

Prof. Pawel Swietach
University of Oxford
Department of Physiology, Anatomy & Genetics
Parks Road
Oxford OX1 3PT
United Kingdom

Dear Prof. Swietach,

Thank you for submitting your manuscript entitled "Acidosis attenuates the hypoxic stabilization of HIF-1 α by activating its lysosomal degradation". The manuscript was assessed by expert reviewers, whose comments are appended to this letter. We invite you to submit a revision if you can address the reviewers' key concerns, as outlined here.

You will see that the reviewers have mixed opinions on the significance and novelty of your paper, while many of their experimental concerns are similar. Editorially we find that your study provides interesting insight into the relationship between acidosis and hypoxia and as such a suitably revised study that completely addresses all of the reviewers' technical concerns is suitable for JCB.

GENERAL GUIDELINES:

Text limits: Character count for an Article is < 40,000, not including spaces. Count includes title page, abstract, introduction, results, discussion, and acknowledgments. Count does not include materials and methods, figure legends, references, tables, or supplemental legends.

Figures: Articles may have up to 10 main text figures. Figures must be prepared according to the policies outlined in our Instructions to Authors, under Data Presentation, <https://jcb.rupress.org/site/misc/ifora.xhtml>. All figures in accepted manuscripts will be screened prior to publication.

Supplemental information: There are strict limits on the allowable amount of supplemental data. Articles may have up to 5 supplemental figures. Up to 10 supplemental videos or flash animations are allowed. A summary of all supplemental material should appear at the end of the Materials and methods section.

Please note that JCB now requires authors to submit Source Data used to generate figures containing gels and Western blots with all revised manuscripts. This Source Data consists of fully uncropped and unprocessed images for each gel/blot displayed in the main and supplemental figures. Since your paper includes cropped gel and/or blot images, please be sure to provide one Source Data file for each figure that contains gels and/or blots along with your revised manuscript files. File names for Source Data figures should be alphanumeric without any spaces or special characters (i.e., SourceDataF#, where F# refers to the associated main figure number or SourceDataFS# for those associated with Supplementary figures). The lanes of the gels/blots should be labeled as they are in the associated figure, the place where cropping was applied should be marked (with a box), and molecular weight/size standards should be labeled wherever possible. Source Data files will be made available to reviewers during evaluation of revised manuscripts and, if your paper is eventually published in JCB, the files will be directly linked to specific figures in the published article.

The typical timeframe for revisions is three to four months. If you anticipate any difficulties in meeting this aforementioned revision time limit, please contact us and we can work with you to find an appropriate time frame for resubmission. Please note that papers are generally considered through only one revision cycle, so any revised manuscript will likely be either accepted or

rejected.

Thank you for this interesting contribution to Journal of Cell Biology. You can contact us at the journal office with any questions at cellbio@rockefeller.edu.

Sincerely,

Johan Auwerx, MD, PhD
Monitoring Editor

Andrea L. Marat, PhD
Deputy Editor

Journal of Cell Biology

Reviewer #1 (Comments to the Authors (Required)):

The authors present a series of in vitro cell assays supporting the claim that the activation of HIF-1 α under low oxygen conditions is influenced by extracellular pH (pHe) (ie, downmodulated upon extracellular acidification). They suggest that this interaction is crucial in preventing excessive fermentation, which could otherwise lead to an overload of acidity in the tumor microenvironment. The reduction in both fermentative but also respiratory fluxes in some cancer cells under acidotic hypoxia (resulting in a shortage of biosynthetic intermediates) is consistent with the observed activation of lysosome-based autophagy.

This study holds significant importance. Specifically, it suggests that hypoxic areas may be underestimated when based on HIF signaling or HRE-dependent hypoxia marker detection. While it is becoming clear in the field that acidosis can occur in moderately oxygenated tumor areas, these data now indicate that acidosis and hypoxia can coexist but not always be detected simultaneously. This has profound implications for radiotherapy, where O₂ levels -detected based on low expression of HIF targets- are used to predict tumor radiosensitivity.

Major comments:

While the selection of C99 and SW1222, as highlighted in Figure 1, is logical, this leads to introduce a bias in the authors's conclusions. What is the rationale for the distinct behavior of these two cell lines? Are specific genetic alterations involved? Furthermore, confusion arises when the HDC111 cell line is used in Figure 2D to show similar behavior to C99 and SW122, while in Figure 1, HCC111 cells show a drop in pHe without a drop in O₂. More broadly, the manuscript's conclusions are drawn without considering the initial selection of cell lines, which show an interesting phenotype but are not representative of most cell lines (i.e., no concomitant reduction in pHe and pO₂). These issues should be addressed to provide readers with a more coherent flow in the presentation of results. Related to this point is the use of medium acidification as a proxy for fermentation. Time-dependent lactate production should be tracked in parallel (and presented in a Figure 1 panel) to support the phenotype and rule out (large) contribution of respiration-induced H⁺ release.

While HIF-1 downmodulation is clearly shown in Figure 2, the reduction in the expression of HRE-dependent targets is less obvious in C99 cells for CA9 and LDHA (not significant for the latter) in Figure 3. The long half-life of CA9 could explain the delayed downmodulation of this protein (which the authors have extensively studied in the past), but then why would the downmodulation be more rapid in SW122? This should be addressed. As for LDHA, measuring extracellular lactate could help the authors strengthen their claims in Figures 2 and 3.

The lack of proteasomal involvement requires further experiments. Additional studies under hypoxia and an evaluation of other proteasomal targets should be presented in addition to the panel in Figure 7C.

Similarly, the involvement of mTOR needs more substantial evidence. While the data are clear for SW1222, this is not the case for CA99 (with no statistical significance for the pHe/pO₂ double variable). Additional reporters should be provided.

The use of DMSO as a vehicle is often unavoidable but can sometimes yield unexpected results. A supplementary figure should compare the effect of DMSO vs. no vehicle for key figures where DMSO was used as a control for DMOG.

Minor comments:

DMOG should also be used in C99 experiments in Figure 3.

The authors did a great job explaining complex experimental setups. However, more details are needed to understand certain figure panels:

-Figure 1A: The acid sensitivity (top left) and resistance (bottom right) require experienced eyes. Superimposing the curves or providing color-based references would help support the claim.

-Figure 1C: It should be mentioned if, after 48 hours at pH 6.4, cells are returned to pH 7.4.

-Figure 3E: The bar graph should present data in the same order as the blots on the left.

-Figure 7C: This figure is referred to as Figure 6C on page 9, which should be corrected.

Finally, the term "alkalosis" typically refers to a blood pH above 7.4. In this study, describing pH 7.4 as "neutral" might be more appropriate.

Reviewer #2 (Comments to the Authors (Required)):

This study by White et al. examines interplay between hypoxia and acidosis in HIF-1a regulation. The authors present several interesting findings, namely that HIF-1a stability under hypoxia is pH-dependent. This raises a new consideration in the field: cells under extended hypoxia may eventually downregulate HIF-1a to prevent acid overload caused by associated fermentation. In general, there are some important discoveries here that can move the field forward. However, there are some internal inconsistencies and points that require further development for the manuscript to be suitable for publication in JCB. I would like to highlight 4 major points that require significant attention:

Major points

1. Figure 3 looks at HIF target genes to see if acidosis under hypoxia attenuates their heightened expression. Some examples are not compelling; most notably, LDHA in the top panel of 3A. Also, the generalities are not well supported. In 4A,B, the authors report on just a single protein (CEACAM6) that was unaffected and use this as evidence that the antagonistic effects of hypoxia and acidosis are not a more general phenomenon. This seems like a missed opportunity to do more global proteomics to see what is generally attenuated or not. The authors even state: "The hypoxic responses of ensemble pathways are not trivial to predict from individual protein changes". Thus, it is hard to deduce larger patterns just purely on these few examples, and the expression analysis would benefit from being expanded.

2. The timing of HIF-1a downregulation in response to acidity in Figure 6 is still unclear. 6B shows that the decay starts before 16h (the band at 16h is less than at 4h), whereas the text mentions that this decay happens "between" 16 and 48h. Moreover, Figure 6B and 6D seem to be conflicting - in 6B, the downregulation starts before hour 16, whereas the conclusion from 6D was that "there was no apparent pH-sensitivity of HIF-1 α at 16 h of hypoxic incubation." These discrepancies need to be corrected to reflect a consistent decay pattern.

3. Figure 7D is a major result of the paper, with the conclusion being that bafilomycin prevents HIF-1a decay under hypoxia. However, the data are not convincing. The representative blot is confusing. In the bafilomycin lanes, the HIF-1a band at 1% O₂ pH 6.4 is smaller than corresponding band at pH 7.4, but the conclusion given is that there is no decrease in level (maybe because the actin control also decreases, but it does not run cleanly). The data points in the graph for hypoxia + bafilomycin are also widely varying; it is no surprise these would not be deemed different. More robust data is required to definitively make this conclusion. In addition to redoing the presented experiment, the authors should test additional lysosome inhibitors and/or lysosome gene knockdowns (especially since TFEB knockdown in 8C has no apparent effect).

4. The lysosomes shown at pH 6.4 under 1% O₂ in Figure 8A seem abnormal/enlarged. The authors quantify lysosome number; is lysosome size also increased? Sometimes, an increase to lysosome size indicates that these organelles are not degrading cargo efficiently and that instead nondigested cargo is building up inside. Is this the case here? While the TORC data in Figure 9 are nice, there is limited data indicating the actual level of degradative flux in these cells. The authors look at LC3 with bafilomycin in 9B, but important controls in this experiment are missing. Also, why is the LC3-II band not seen for 1% O₂ pH 6.4 in 9B but it is the strongest band in 9A? The authors need to more conclusively show that lysosomes are in fact more active/digestive in their tested scenario. Several potential alternative approaches are given in a seminal review on guidelines for monitoring autophagy: doi.org/10.1080/15548627.2020.1797280

Minor points

1. Labeling in the graphs of Figure 8C is off.

Reviewer #3 (Comments to the Authors (Required)):

In this paper, the authors attempt to uncouple the effects of cellular acidosis and oxygen depletion on HIF-1 signaling using colorectal cancer cell lines that survive acidic conditions. By controlling the pH and oxygen content of cell culture medium, the authors studied the change of HIF-1 α content under conditions with different oxygen and pH treatment combinations. They concluded that HIF-1 α induction, as well as its representative targets (CA9, LDHA, PDK1) dose-dependently decreased with concurrent reductions in extracellular pH. Furthermore, by applying MG-132 and BafA1, which targets proteasomes and lysosomes respectively, they proposed that reduction of HIF-1 α expression is mediated by the lysosomes but not the proteasomes. Finally, they provided some evidence showing that acidotic hypoxia increased the abundance of lysosomes and activated autophagy by disabling the inhibitory effect of mTORC1 signaling, resulting in HIF-1 α degradation. While the finding that HIF-1 α expression decreased with the reductions in extracellular pH seems interesting, the data presented regarding to the regulatory mechanism of this process were rather preliminary and of low quality. Given the fact that a number of reports have studied the interaction between acidosis and HIF (Parks et al., 2013; Selfridge et al., 2016; Tang et al., 2012; Willam et al., 2006), the novelty of the current manuscript is not so strong. In addition, the paper is too much focused on the correlation analysis and would thus need more mechanistic work to strengthen the conclusions, as specified below:

Major comments:

1. It is unclear how general the reduction of HIF-1 α expression decreased with the reductions in extracellular pH will occur, as this phenomenon is actually opposite to some of the previous findings (e.g., PMID: 15181450; PMID: 27488520). More importantly, the authors did not yet explore the functional readouts of this process, for example, would the cells more susceptible to cell death if this process is blocked?
2. The acidic conditions set by varying [HCO₃⁻], are still rather artificial, though better than using non-volatile buffers. The authors may also want to explore whether the natural end product of fermentation, the lactic acid, could be used as a regulator of HIF-1 α expression under hypoxia condition, or any physiological condition can be applied in their context?
3. In Figure 7C, the treatment of MG132 in C99 or SW122 cells should also be conducted under acidic hypoxia conditions (pHe6.4, 1% O₂). In addition, another positive control, in addition to Hif1a, regulated by the proteasomes is also suggested to be included.
4. The imaging results to conclude that hypoxia significantly increased the lysosomal number (Fig. 8 and S4), is of poor quality and unconvincing, more related evidences (e.g. the LAMP1/2 expression, the Cathepsin content/activity) should be provided.
5. The data shown in Figure 9 are rather correlative findings, the author should also test whether inhibitors of autophagy (e.g. 3-MA) and the mTORC1 (rapamycin/Torin1) may regulates the HIF-1 α expression under hypoxia condition?
6. For most of the western blot experiments with three biological replicates, all results for the three replicates should ideally be shown, especially considering the high Standard Deviation presented in some of these data (e.g., Fig. 4, Fig. 5C). If outliers with unusual values exists, for example in Fig. 5C, an outlier with unusual low value seems existing in each of the experimental groups. A repeat of such kind of experiment should be conducted.

Minor comments:

1. In Fig. 2a-d, and Fig. 3b,3d: the labels for pH values are incomplete.
2. Fig. 8C: the "siScr" and "siTFEB" in the TFEB/ β -actin and HIF-1 α / β -actin quantification charts should be clearly written.

Reviewer #1:

Thank you for your helpful comments and constructive suggestions on how to improve our manuscript. Your comments are copied below in **this typeface** and are followed by our response. Changes to the manuscript in response to your comments have been highlighted in **blue font**.

The authors present a series of in vitro cell assays supporting the claim that the activation of HIF-1 α under low oxygen conditions is influenced by extracellular pH (pHe) (ie, downmodulated upon extracellular acidification). They suggest that this interaction is crucial in preventing excessive fermentation, which could otherwise lead to an overload of acidity in the tumor microenvironment. The reduction in both fermentative but also respiratory fluxes in some cancer cells under acidotic hypoxia (resulting in a shortage of biosynthetic intermediates) is consistent with the observed activation of lysosome-based autophagy. This study holds significant importance. Specifically, it suggests that hypoxic areas may be underestimated when based on HIF signaling or HRE-dependent hypoxia marker detection. While it is becoming clear in the field that acidosis can occur in moderately oxygenated tumor areas, these data now indicate that acidosis and hypoxia can coexist but not always be detected simultaneously. This has profound implications for radiotherapy, where O₂ levels -detected based on low expression of HIF targets- are used to predict tumor radiosensitivity.

Thank you; we are delighted that you recognise the significance and implications of our findings.

While the selection of C99 and SW1222, as highlighted in Figure 1, is logical, this leads to introduce a bias in the authors's conclusions. What is the rationale for the distinct behavior of these two cell lines? Are specific genetic alterations involved? Furthermore, confusion arises when the HDC111 cell line is used in Figure 2D to show similar behavior to C99 and SW1222, while in Figure 1, HDC111 cells show a drop in pHe without a drop in O₂. More broadly, the manuscript's conclusions are drawn without considering the initial selection of cell lines, which show an interesting phenotype but are not representative of most cell lines (i.e., no concomitant reduction in pHe and pO₂). These issues should be addressed to provide readers with a more coherent flow in the presentation of results.

We have now revised the *Introduction* to better explain the rationale for selecting cell lines. Briefly, we started with a panel of 68 CRC lines that covers the range of phenotypes found in human cancers, and narrowed this down to specific lines that are most compatible with our study objectives. Since our primary aim was to study the interaction between hypoxia and acidosis, we considered two criteria: (i) suitable CRC lines must survive at low medium pH to manifest a hypoxic interaction (i.e., of acid-resistant phenotype), and (ii) deplete oxygen to produce hypoxia. This combination was satisfied by C99 and SW1222 cells, which progressed as the principal lines for the study (Fig 1A for condition (i), Fig 1C for condition (ii)). By starting with SW1222/C99 cells, our results gave confidence in the physiological relevance of the attenuation of HIF-1 α stabilization at low extracellular pH. To test the extent to which our conclusions could be generalised, we considered additional cell lines that partially met the criteria, e.g., HT29 and HDC111 which have intermediate acid-sensitivity and do not substantially deplete medium O₂. As **new data**, we verify the pHe/pO₂ interaction on HIF-1 α in HT29 cells (Fig 2E). By expanding to other cell lines, we showed that this action is a more general phenomenon. For example, HDC111 meets condition (i) but not (ii), yet still manifests the interplay between acidosis and hypoxia on HIF-1 α (Fig 2D). Overall, our mechanism is likely to be general, provided the cells survive acidic conditions. For a more coherent flow of the manuscript, we have now clarified this rationale for additionally measuring HIF-1 α in HDC111 and HT29 cells within the main text.

Previously, we showed (Michl *Cell Rep*, 2022) that genes of mitochondrial respiration, notably *NDUFS1*, are essential for surviving low environmental pH because glycolysis is strongly suppressed at low pH (Blaszczak *BJC*, 2022). Thus, the nature of acid-resistance lies in the ability of cells to support mitochondrial respiration, even at low pH. We now confirm this for C99 and SW1222. In terms of understanding molecular and genetic underpinnings, we previously demonstrated (Michl *PNAS*, 2024) that high *CEACAM5* or *CEACAM6* expression correlates with acid-resistant phenotypes across the panel of 68 CRC cell lines. In the same study, we also identified a weak but significant correlation between mutations in *KRAS* and acid resistance, but *KRAS* driver mutations were not exclusive to or absolutely required for an acid-resistant phenotype. Illustrative of the complex and multi-factorial nature of acid-sensitivity, SW1222 cells carry a *KRAS* mutation, although *KRAS* is thought to be wildtype in C99 cells (Medico *Nat Comms*, 2015). Critically, repositories such as DepMap (<https://depmap.org/>) provide sequences for only 40 of the 68 CRC cells we have phenotyped, and include only 2 of the most acid-sensitive lines. This lack of information across the breadth of acid-sensitivity reduces the statistical power to discover correlations between phenotype and mutations. A comprehensive study is warranted, once the full range of CRC lines is sequenced. We hope R1 agrees that further molecular understanding of acid-resistance mechanisms would fall outside the scope of this paper.

Related to this point is the use of medium acidification as a proxy for fermentation. Time-dependent lactate production should be tracked in parallel (and presented in a Figure 1 panel) to support the phenotype and rule out (large) contribution of respiration-induced H⁺ release.

We have validated our acidification assay as a surrogate of lactic acid production using pancreatic ductal adenocarcinoma cell lines in Blaszczak *Brit J Cancer*, 2022. Nonetheless, it's a good idea to confirm in CRC, which we present as **new data** in Fig 1D. Experiments were under all combinations of conditions (pHe, HIF stabilization by DMOG) and performed in HDC111, HT29, C99, and SW1222 cells, to cover cells of distinct metabolic phenotypes (as show in Fig 1C). Acidification assays were terminated at 8 or 17 h. The time-course of HPTS fluorescence reported cumulative

H⁺ production, which was paired with an end-point [lactate] biochemical assay. These measurements correlated with a slope of 1, verifying that H⁺ production under our conditions is primarily glycolytic. This is consistent with lactic acid being involatile, unlike CO₂, which can escape and therefore has a lesser impact on extracellular acidification. The same principle is the basis for the Seahorse analyser – our method is a more versatile adaptation of the device (Blaszczak, *STAR Methods*, 2024).

While HIF-1 downmodulation is clearly shown in Figure 2, the reduction in the expression of HRE-dependent targets is less obvious in C99 cells for CA9 and LDHA (not significant for the latter) in Figure 3. The long half-life of CA9 could explain the delayed downmodulation of this protein (which the authors have extensively studied in the past), but then why would the downmodulation be more rapid in SW122? This should be addressed.

There is, perhaps, a misunderstanding. We are unclear as to what is meant by “the downmodulation [of CA9] being more rapid in SW122?”. The patterns of CA9 expression measured at 48 h hypoxia (1% O₂) are comparable between C99 (Fig 4A) and SW1222 cells (Fig S2A). When considering quantification of these immunoblots, CA9 downmodulation under acidotic hypoxia (relative to alkalotic hypoxia) is approximately 35-55% in both cell lines, and there is a significant pHe:pO₂ interaction for CA9 levels in both cell lines. Indeed, LDHA has a less prominent effect, but we have now added **new proteomic data** to classify protein abundance responses according to the strength of their acid-hypoxia interaction (Fig 3). We find 59 proteins correlating significantly with HIF-1α levels, i.e. less abundant in acidotic hypoxia relative to alkalotic hypoxia. Other canonical HRE-dependent targets such as also PDK1 and CA9 followed HIF-1α closely.

As for LDHA, measuring extracellular lactate could help the authors strengthen their claims in Figures 2 and 3.

Our measurements of fermentative rate and **new data** on the concordance between H⁺ and lactate production make a case that metabolic-rate responses to acidosis and hypoxia agree with HIF target responses. We show that DMOG increases acid production when the pre-treatment pHe is 7.4, but this effect is ablated when the pre-treatment pHe was reduced to 6.4 (Fig 5A). We have now validated that acid production reflects lactic acid release (Fig 1D).

The lack of proteasomal involvement requires further experiments. Additional studies under hypoxia and an evaluation of other proteasomal targets should be presented in addition to the panel in Figure 7C.

This is a fair point, which we address with **new data**. Firstly, we directly measured proteasomal activity using a luciferase-based assay (Fig 8H) in C99, SW1222, HT29 and HDC111 cells at pHe 7.4, 6.9, or 6.4, with or without DMOG treatment. Neither pHe nor DMOG had a significant effect on proteasomal activity. Moreover, there was no significant interaction between pHe and DMOG on proteasomal activity. This provides further support that, relative to alkaline conditions, acidosis does not induce an increase in proteasomal activity that could be responsible for the observed HIF-1α degradation. In addition, we provide **new data** (Fig 8F-G) showing that HIF-1α induction *remains* blunted under acidotic hypoxia, relative to alkalotic hypoxia, even when the proteasome is inhibited by MG-132/epoxomicin. We include ubiquitin to demonstrate the extent to which MG-132 and epoxomicin inhibit proteasomal activity (MG-132>epoxomicin).

Similarly, the involvement of mTOR needs more substantial evidence. While the data are clear for SW1222, this is not the case for CA99 (with no statistical significance for the pHe/pO₂ double variable). Additional reporters should be provided.

We draw attention to the significant pHe:pO₂ interaction term for Ser240/244 phosphorylation of S6 in C99 cells (Fig 10E). Hypoxia, independently of acidosis, reduced S6 phosphorylation but there was a *further* reduction when both stimuli were presented concurrently. This is consistent with inhibition of mTORC1 signaling under acidotic hypoxia, relative to alkalotic hypoxia. Whilst Thr389-phosphorylation of S6K did not have a significant pHe:pO₂ interaction term in C99 cells, there was still a clear trend towards lower phosphorylation levels under acidotic hypoxia relative to alkalotic hypoxia. In SW1222 cells, the pHe:pO₂ interaction term emerged as significant for S6K phosphorylation but not S6 phosphorylation, but there was a clear trend of lowest S6 phosphorylation levels under acidotic hypoxia. To clarify our interpretation of S6K/S6 phosphorylation patterns reflecting a lower mTORC1 activity in acidotic hypoxia relative to alkalotic hypoxia, we provide **new data** (Fig. S5C). We use Ser65-phosphorylation of 4E-BP1 because it is an alternative mTORC1 substrate to S6K (unlike S6, which is phosphorylated by S6K) (Böhm *Mol Cell*, 2021). In SW1222 cells, 4E-BP1 phosphorylation was ablated by acidotic hypoxia but not acidosis or hypoxia alone, a finding consistent with the S6K/S6 phosphorylation patterns. In C99 cells, 4E-BP1 phosphorylation could not be harnessed as a readout of mTORC1 activity because baseline 4E-BP1 levels were extremely low in acidotic hypoxia as shown right.

The use of DMSO as a vehicle is often unavoidable but can sometimes yield unexpected results. A supplementary figure should compare the effect of DMSO vs. no vehicle for key figures where DMSO was used as a control for DMOG.

We provide **new data** (Fig S2C) to confirm that DMSO *per se* does not alter readouts of HIF signaling (such as CA9 and LDHA expression), relative to conditions of no DMSO at 21% O₂.

DMOG should also be used in C99 experiments in Figure 3.

This figure is now Fig 4. We address this with **new data** (Fig S2B): immunoblots for CA9 and LDHA in C99 cells following 48 h treatment at pHe 7.4 or 6.4 and either DMSO or 1 mM DMOG. Consistent with the hypoxic result for C99, acidic pHe inhibits CA9, but not LDHA induction, under DMOG.

Figure 1A: The acid sensitivity (top left) and resistance (bottom right) require experienced eyes. Superimposing the curves or providing color-based references would help support the claim.

We have modified the panel so data on all cell lines fall in a row, showing the progressive shift towards more acid-resistant phenotypes from the position of the black curve, relative to grey curves (all lines).

Figure 1C: It should be mentioned if, after 48 hours at pH 6.4, cells are returned to pH 7.4.

Thank you for bringing this to our attention. We now clarify, in the Fig 1B-C legend, that pre-treatments at pHe 7.4 or 6.4 lasted for 48 h, after which cells were returned to medium of pH 7.4 immediately prior to metabolic measurements.

Figure 3E: The bar graph should present data in the same order as the blots on the left. Figure 7C: This figure is referred to as Figure 6C on page 9, which should be corrected.

Amended as requested. (Fig 3E now appears as Fig 4E).

Finally, the term "alkalosis" typically refers to a blood pH above 7.4. In this study, describing pH 7.4 as "neutral" might be more appropriate.

We acknowledge that normal arterial pH falls within the range 7.35-7.45 and, in clinical situations, an alkalosis refers to an arterial pH greater than 7.45. However, neutral pH is technically 7.0 and we would wish to avoid incorrectly labelling alkaline cell culture conditions of 7.4 as 'neutral' (indeed, condition pH 6.9 is closest to neutral).

Reviewer #2:

Thank you for your helpful comments and constructive suggestions on how to improve our manuscript. Your comments are copied below in **this typeface**, and are followed by our response. Changes to the manuscript in response to your comments have been highlighted in **blue font**.

This study by White et al. examines interplay between hypoxia and acidosis in HIF-1 α regulation. The authors present several interesting findings, namely that HIF-1 α stability under hypoxia is pH-dependent. This raises a new consideration in the field: cells under extended hypoxia may eventually downregulate HIF-1 α to prevent acid overload caused by associated fermentation. In general, there are some important discoveries here that can move the field forward. However, there are some internal inconsistencies and points that require further development for the manuscript to be suitable for publication in JCB. I would like to highlight 4 major points that require significant attention:

Thank you for these comments and highlighting issues, which we addressed with extensive new data.

Figure 3 looks at HIF target genes to see if acidosis under hypoxia attenuates their heightened expression. Some examples are not compelling; most notably, LDHA in the top panel of 3A. Also, the generalities are not well supported. In 4A,B, the authors report on just a single protein (CEACAM6) that was unaffected and use this as evidence that the antagonistic effects of hypoxia and acidosis are not a more general phenomenon. This seems like a missed opportunity to do more global proteomics to see what is generally attenuated or not. The authors even state: "The hypoxic responses of ensemble pathways are not trivial to predict from individual protein changes". Thus, it is hard to deduce larger patterns just purely on these few examples, and the expression analysis would benefit from being expanded.

We agree that the significance of HIF merits a proteomic investigation, which we undertook using a new mass spec machine installed in Oxford in January 2025 (the first project assigned to the device). We were able to resolve >9,000 peptides and identify changes in abundance under acidosis, hypoxia and their combination. These **new data** are presented throughout the manuscript. In Fig 3C-D, we confirm the acid/hypoxia interaction of HIF-1 α and identify 59 proteins correlating significantly with HIF-1 α levels. Other canonical HRE-dependent targets such as also PDK1 and CA9 followed HIF-1 α closely. LDHA was shown to respond with weaker correlation. We also identified proteins that do not follow the HIF1 α -trend, including: (i) protein responses showing synergy between hypoxia and acidosis (e.g. CEACAM5), and (ii) proteins that were acid-induced but hypoxia-insensitive (e.g. AKR1C2). In Fig 3E-F, we further analyse responses to acidosis and hypoxia, dividing this into 4 distinct behaviours. These findings indicate that the hypoxia/acidosis interaction for HIF-1 α and several of its targets is *not* a general run-down of proteins, but a specific effect. We found enrichment of proteins with short half-life (<8 h) among downregulated proteins under the combination of acidosis and hypoxia (but not acidosis or hypoxia alone), indicating at least some role for a stimulated degradative pathway for regulating transient proteins, such as HIF-1 α . In Fig 7, we use proteomics to indicate lysosomal mechanisms. This analysis sets the scene for more mechanistic studies that follow.

The timing of HIF-1 α downregulation in response to acidity in Figure 6 is still unclear. 6B shows that the decay starts before 16h (the band at 16h is less than at 4h), whereas the text mentions that this decay happens "between" 16 and 48h. Moreover, Figure 6B and 6D seem to be conflicting - in 6B, the downregulation starts before hour 16, whereas the conclusion from 6D was that "there was no apparent pHe-sensitivity of HIF-1 α at 16 h of hypoxic incubation." These discrepancies need to be corrected to reflect a consistent decay pattern.

We believe the results are consistent, when overall changes are considered. Under acidosis and hypoxia imposed at the same time (Fig 6B), HIF-1 α levels decrease after the initial 4 h peak, but the statistically significant change is between 16 and 48 hr. Under alkaline conditions (Fig 6A), there is no statistically significant decrease in HIF-1 α beyond 4 hr of treatment. Thus, 16 h is the latest (measured) time-point at which we do not expect a significant difference in HIF-1 α levels in acidotic versus alkalotic hypoxia. To test this, we probed HIF-1 α under alkalotic hypoxia and acidotic hypoxia on the same immunoblot membrane in Fig 6D, to allow for a direct comparison of abundance. Indeed, Fig 6D shows the pHe-insensitivity of hypoxic HIF-1 α induction at 16 h. We have revised the text to reflect this.

Figure 7D is a major result of the paper, with the conclusion being that bafilomycin prevents HIF-1 α decay under hypoxia. However, the data are not convincing. The representative blot is confusing. In the bafilomycin lanes, the HIF-1 α band at 1% O₂ pHe 6.4 is smaller than corresponding band at pHe 7.4, but the conclusion given is that there is no decrease in level (maybe because the actin control also decreases, but it does not run cleanly). The data points in the graph for hypoxia + bafilomycin are also widely varying; it is no surprise these would not be deemed different. More robust data is required to definitively make this conclusion.

We appreciate this is an important test of our mechanism and agree that the spread of densitometric quantification may give the impression of committing a type II error, but our statistics are batch-corrected and much of the variability arises from between-gel differences. Given the spread on quantification, we now instead show all blots across Fig 8 and Fig S3, including **new repeats**. If a degradative process is activated at low pH, we believe the best test is to determine if an inhibitor collapses the effect of pH under hypoxia, i.e. testing "lanes" 7 and 8. In all our blots, we show that the contrast between the two lanes decreases with bafilomycin. We interpret this as a rescue. We note that these experiments are challenging: growth under the combination of acidity and hypoxia is restricted and adding bafilomycin only exacerbates this. We note that the contrast between lanes 7/8 also collapses with 3-MA, another inhibitor of autophagy, to complement the bafilomycin result. To justify our experiments on the lysosomal pathway, we now present **new**

proteomic data, shown in Fig 3, that describe the effect of acidosis, hypoxia and their interaction. It confirms that hypoxic stabilization of HIF-1 α is attenuated at low pHe, which reflects changes in a subset of downstream targets. This analysis shows that our downregulation of HIF-1 α under acidotic hypoxia is not a manifestation of overall rundown, as some proteins respond synergistically to acidosis and hypoxia. Enrichment analysis shows that downregulated proteins are more typically short-lived, indicating that a degradative process has at least some role. We eliminate proteasomal mechanisms (dually, by showing this is pVHL-insensitive and using an array of blockers: MG132 and epoxomicin). We trust the volume of additional data provides convincing evidence for a mechanistic role of lysosomes in degrading HIF-1 α under acidotic hypoxia.

In addition to redoing the presented experiment, the authors should test additional lysosome inhibitors and/or lysosome gene knockdowns (especially since TFEB knockdown in 8C has no apparent effect).

The concern with knockdown/knockout of is that this may instigate adaptive or compensatory responses, therefore exposing selected phenotypes, rather than mechanisms. We reason that a more informative interrogation of an acutely activated signaling response is through pharmacology. Unfortunately, many canonical lysosome inhibitors – often based on pH-sensitive structures or with weakly acidic/basic moieties – do not function properly in media of low pHe, as we have previously shown for chloroquine (Michl, *Cell Rep*, 2023). Reviewer R3 suggested we use 3-MA, which we now include as **new data** (Fig S5). This shows that 3-MA, an inhibitor of autophagosome formation (i.e. upstream to lysosomes in the macro-autophagy pathway), renders hypoxic HIF-1 α stabilization and CA9 induction less pHe-sensitive. This result is consistent with 3-MA sharing the same pathway that is blocked under acidotic but not alkalotic hypoxia.

The lysosomes shown at pHe 6.4 under 1% O₂ in Figure 8A seem abnormal/enlarged. The authors quantify lysosome number; is lysosome size also increased? Sometimes, an increase to lysosome size indicates that these organelles are not degrading cargo efficiently and that instead nondigested cargo is building up inside. Is this the case here?

We have now quantified the diameter of circular LysoBrite-positive particles identified as lysosomes in Fig 9A. Particle analysis used the circle Hough Transform with a degree of tolerance on circularity (sensitivity parameter 1.0), performed against high-resolution images (2048x2048, 16-bit images). The algorithm was user-trained to recognise lysosomes using a subset of images. In this unsupervised analysis, **new analyses** find that acidotic hypoxia does not affect the size of identified lysosomes (Fig 9C). Further analysis indicated that acidotic hypoxia shifted radially the distribution of identified lysosomes away from the nucleus (Fig 9D). The distinct appearance of lysosomes under acidotic hypoxia may be explained by the emergence of cytoplasmic regions with higher lysosomal density due to number and re-distribution. For further evidence that the lysosomal degradative activity remains intact, we present **new data** (Fig 9F): we used imaging with the cathepsin B substrate “Magic Red Substrate MR-(RR₂)” to show that the greatest accumulation of its fluorescent product of degradation occurs with acidotic hypoxia. Consistently, we find that cleavage of pro-cathepsin B to active cathepsin B is greater in acidotic hypoxia, relative to alkalotic hypoxia.

While the TORC data in Figure 9 are nice, there is limited data indicating the actual level of degradative flux in these cells. The authors look at LC3 with bafilomycin in 9B, but important controls in this experiment are missing. Also, why is the LC3-II band not seen for 1% O₂ pHe 6.4 in 9B but it is the strongest band in 9A? The authors need to more conclusively show that lysosomes are in fact more active/digestive in their tested scenario. Several potential alternative approaches are given in a seminal review on guidelines for monitoring autophagy: doi.org/10.1080/15548627.2020.1797280

Thank you for these comments. We now include control bafilomycin A1-free lanes for all the pHe/pO₂ combinations tested in Fig. 10B. To clarify, bafilomycin A1 was used to inhibit the fusion of lysosomes to autophagosomes, thus preventing the degradation of LC3-II localised to the internal autophagosome membrane (Sharifi *Methods Mol Biol*, 2015). Consequently, LC3-II levels measured in the presence of bafilomycin A1 confirmed the effect of acidotic hypoxia specifically on autophagosome formation. Regardless of pHe/pO₂, bafilomycin A1 increased LC3-II abundance to such an extent that the immunoblot membrane became saturated well before any LC3-II became detectable in control (bafilomycin A1-free) lanes. We can show this for longer membrane development times: LC3-II did eventually become detectable in these bafilomycin A1-free lanes. This explains the difference between Fig 10A and Fig 10B. An example of such an over-exposed membrane is shown on the right. To present an immunoblot in Fig 10B that encompasses all conditions, images were selected from a timepoint in membrane development where the bafilomycin A1 lanes were non-saturating. This was necessarily short and not sufficient to make LC3-II visible in the bafilomycin A1-free lanes. In Fig 10A, we present bafilomycin A1-free membranes that were allowed to develop until near-saturating levels of LC3-II could be visualised. Further evidence for increased lysosomal activity is presented with new data in response to the previous point.

Labeling in the graphs of Figure 8C is off.

We checked and corrected any issues in Fig 8C, now appearing as Fig 10A.

Reviewer #3:

Thank you for your helpful comments and constructive instructions on how to improve our manuscript. Your comments are copied below in **this typeface**, and are followed by our response. Changes to the manuscript in response to your comments have been highlighted in **blue font**.

In this paper, the authors attempt to uncouple the effects of cellular acidosis and oxygen depletion on HIF-1 signaling using colorectal cancer cell lines that survive acidic conditions. By controlling the pH and oxygen content of cell culture medium, the authors studied the change of HIF-1 α content under conditions with different oxygen and pH treatment combinations. They concluded that HIF-1 α induction, as well as its representative targets (CA9, LDHA, PDK1) dose-dependently decreased with concurrent reductions in extracellular pH. Furthermore, by applying MG-132 and BafA1, which targets proteasomes and lysosomes respectively, they proposed that reduction of HIF-1 α expression is mediated by the lysosomes but not the proteasomes. Finally, they provided some evidence showing that acidotic hypoxia increased the abundance of lysosomes and activated autophagy by disabling the inhibitory effect of mTORC1 signaling, resulting in HIF-1 α degradation. While the finding that HIF-1 α expression decreased with the reductions in extracellular pH seems interesting, the data presented regarding to the regulatory mechanism of this process were rather preliminary and of low quality. Given the fact that a number of reports have studied the interaction between acidosis and HIF (Parks et al., 2013; Selfridge et al., 2016; Tang et al., 2012; Willam et al., 2006), the novelty of the current manuscript is not so strong. In addition, the paper is too much focused on the correlation analysis and would thus need more mechanistic work to strengthen the conclusions, as specified below:

Thank you for these comments. Below, we outline our efforts to improve the mechanistic insight with **new data**. Whilst we appreciate others have embarked on this topic, their outcomes have been variable. Our study attempts to address this through a well-controlled and systematic approach, starting from choice of cell lines to superior pH control, the first to follow good practice in pH control (Michl *Comm Biol*, 2019).

It is unclear how general the reduction of HIF-1 α expression decreased with the reductions in extracellular pH will occur, as this phenomenon is actually opposite to some of the previous findings (e.g., PMID: 15181450; PMID: 27488520). More importantly, the authors did not yet explore the functional readouts of this process, for example, would the cells more susceptible to cell death if this process is blocked?

We now include **new data** to verify the acid-hypoxia interaction in four CRC lines (Fig 2). The robustness of this finding is related to accurate control of pH, which we exercise in our conditions through the use of CO₂/HCO₃⁻ buffering. Findings in PMID 15181450 (Mekhail *Nat Cell Biol*, 2004) and PMID 2748850 (Filatova *Cancer Res*, 2016) both reflect situations where acidosis has been generated by supplementing media with non-volatile buffers followed by pH titration outside an incubator. We have previously shown (Michl *Comm Biol*, 2019) that media prepared in such a way show instability because the pH set outside an incubator is certainly different to the bulk of the cells' experience under 5% CO₂ incubation. As the non-volatile buffer attempts to "defend" pH against CO₂/HCO₃⁻ buffering, pH becomes unstable. We now characterise the pHe-dependent inhibition of hypoxic HIF-1 α responses with **new proteomic data**, shown in Fig 3. We also present **new data** on the functional significance of our acidosis-hypoxia interaction in Fig 5C. These findings relate the effect of pHe on HIF stabilization, with consequences on cell growth over six days. We show how a prior period of full HIF stabilization renders cells vulnerable to a subsequent period in acidosis. We arrive at this conclusion by comparing four protocols that vary DMOG treatment: absent, first two days only, last four days only, all six days of culture. Normoxic incubations were performed over a range of pHe (6.4, 6.9, 7.4). DMOG treatments either matched the normoxic pH, or were performed at pH 7.4 to maximally stabilise HIF. If DMOG treatment had no carry-on effect that exposed a subsequent vulnerability, then overall growth outcomes should be proportional to the number of days in DMOG treatment, irrespective of order. We found that two days of DMOG/pH 7.4 at the start of experiments, followed by normoxic experiments at reduced pH resulted in lower-than-expected growth at the end of the six-day protocol (Fig 5C, green arrows). We explain this in terms of early (unhindered) HIF signaling causing a switch from a respiratory to fermentative phenotype, which becomes a vulnerability under low pHe incubation because this blocks fermentation. This negative effect is attenuated if the prior DMOG treatment was at matching pHe, i.e. to prevent the full extent of HIF-1 α signaling.

The acidic conditions set by varying [HCO₃⁻], are still rather artificial, though better than using non-volatile buffers. The authors may also want to explore whether the natural end product of fermentation, the lactic acid, could be used as a regulator of HIF-1 α expression under hypoxia condition, or any physiological condition can be applied in their context?

This is an excellent idea. We have repeated key experiments with lactic acidosis, i.e. the decrease in [NaHCO₃] was replaced with [NaLactate] rather than [NaCl]. **New data** are shown in Fig 2G. Lactic acidosis, like a "simple" acidosis, attenuates HIF-1 α stabilization under hypoxia. Lactate, independently of H⁺ ions, did not significantly affect hypoxic HIF-1 α induction.

In Figure 7C, the treatment of MG132 in C99 or SW122 cells should also be conducted under acidic hypoxia conditions (pHe6.4, 1% O₂). In addition, another positive control, in addition to Hif1a, regulated by the proteasomes is also suggested to be included

We provide **new data** (Fig S8F-G) showing that *hypoxic* HIF-1 α induction remains blunted under acidic conditions, even when the proteasome is inhibited by either MG-132 or epoxomicin. We include ubiquitin to confirm moderate

proteasomal inhibition by MG-132 and stronger inhibition with epoxomicin. Acid attenuation of hypoxic HIF-1 α stabilization persists even under proteasomal inhibition. We also supply **new data** showing that proteasomal activity is pHe-insensitive (Fig 8H), providing a mechanistic explanation why this degradative pathway would not be activated at low pH. Furthermore, **new proteomic data** (Fig 3) showed an enrichment of short-lived proteins among features downregulated at acidotic hypoxia, pointing to the activation of a degradation pathway. Further analysis (Fig 7) showed an enrichment in lysosomal-pathway proteins under acidotic hypoxia, but not of proteasomal components. Together with multiple lines of evidence showing higher lysosomal activity in acidotic hypoxia (including **new data** in Fig 9F), we make a case for lysosomal mechanisms.

The imaging results to conclude that hypoxia significantly increased the lysosomal number (Fig 8 and S4), is of poor quality and unconvincing, more related evidences (e.g. the LAMP1/2 expression, the Cathepsin content/activity) should be provided.

We respectfully disagree: the confocal imaging (2048x2048 at 16-bit depth) was sufficient for a Hough Transform to identify particles and quantify their size, intensity, and position relative to the nearest nucleus. Training the algorithm using a subset of data determined the criteria for lysosomes (Fig S4B), allowing unsupervised analysis. It is possible that the submission system compresses the images, which is why we now include additional analyses to verify the robustness of our imaging (Fig 9: lysosomal size, distance to nucleus). As mentioned in relation to the previous comment, in support of the LysoBrite images, **new proteomic data** indicate an enrichment of autolysosome-associated proteins selectively under acidotic hypoxia (Fig 7). For additional metrics of lysosomes, we now use imaging (Fig 9F) with the cathepsin B substrate "Magic Red Substrate MR-(RR₂)" to show that the greatest accumulation of its fluorescent product of degradation occurs with acidotic hypoxia. Consistently, we find that pro-cathepsin B cleavage to active cathepsin B is greater in acidotic hypoxia relative to alkalotic hypoxia.

The data shown in Figure 9 are rather correlative findings, the author should also test whether inhibitors of autophagy (e.g. 3-MA) and the mTORC1 (rapamycin/Torin1) may regulate the HIF-1 α expression under hypoxia condition?

We present **new data** (Fig S5) demonstrating the action of 3-MA and rapamycin (as suggested) on HIF-1 α and its reporter CA9. Given that autophagy/lysosomal activities are strongly activated under acidotic hypoxia relative to alkalotic hypoxia, we draw attention to comparisons between the effects of acidotic hypoxia relative to alkalotic hypoxia. In the presence of 3-MA, alkalotic hypoxia produced only mild HIF-1 α stabilization (Fig S5B) which may relate to a number of targets of this drug. Critically, however, 3-MA did not proportionately reduce HIF-1 α stabilization under acidotic hypoxia. Given that 3-MA blocks autophagosome formation, and that these organelles normally emerge under acidotic hypoxia, we interpret our results as a rescue, i.e. blocking the HIF-1 α degradative pathway selectively activated at low pHe/low pO₂. Whilst 3-MA may have multiple actions, we did not observe major changes in mTORC1 readouts (S6K and S6 phosphorylation; Fig S5A), which argues for fewer off-target actions, relative to actions on autophagosome formation. Rapamycin was verified to strongly inhibit mTORC1 signaling (S6K, S6 phosphorylation; Fig S5A). This reduced the extent of HIF-1 α stabilization under alkalotic hypoxia, which is consistent with the role of mTORC1 in the hypoxic pathway. However, rapamycin had no additional effect on HIF-1 α stabilization under acidotic hypoxia. This lack of an additive effect is expected because both acidity and rapamycin inhibit mTORC1. Our new findings are consistent with observations that mTORC1 signaling is already inhibited under acidotic hypoxia (Fig 10D-E; Fig. S5C) thus no longer responsive to rapamycin actions.

For most of the western blot experiments with three biological replicates, all results for the three replicates should ideally be shown, especially considering the high Standard Deviation presented in some of these data (e.g., Fig 4, Fig 5C). If outliers with unusual values exist, for example in Fig 5C, an outlier with unusual low value seems existing in each of the experimental groups. A repeat of such kind of experiment should be conducted.

Where the quantification of immunoblots showed a high standard deviation we provide **new additional repeats**. In particular, 4 immunoblots are now quantified for NDUFS1 and CEACAM6 (as suggested). It would not be feasible to display every single immunoblot quantified within the figure limits, given that there are 29 immunoblot experiments quantified in the main figures alone. However, acknowledging some of the most mechanistically important immunoblots – the modulation of the HIF-1 α pHe/pO₂ interplay by bafilomycin A1 (Fig 8A-B) in C99 and SW1222 cells – we display the immunoblots for a further repeats in Fig S3A-B.

In Fig 2a-d, and Fig 3b,3d: the labels for pH values are incomplete.

We have restructured the figures to make the presentation clearer. Thank you for pointing our inadequacy on this matter.

Fig 8C: the "siScr" and "siTFEB" in the TFEB/ β -actin and HIF-1 α / β -actin quantification charts should be clearly written.

Done – thank you.

May 13, 2025

RE: JCB Manuscript #202409103R

Pawel Swietach
University of Oxford

Dear Prof. Swietach:

Thank you for submitting your revised manuscript entitled "Acidosis attenuates the hypoxic stabilization of HIF-1 α by activating lysosomal degradation". The reviewers now support publication therefore we would be happy to publish your paper in JCB pending final revisions necessary to meet our formatting guidelines (see details below).

A. MANUSCRIPT ORGANIZATION AND FORMATTING:

- 1) Text limits: Character count for Articles is < 40,000, not including spaces. Count includes abstract, introduction, results, discussion, and acknowledgments. Count does not include title page, figure legends, materials and methods, references, tables, or supplemental legends.
- 2) Figures limits: Articles may have up to 10 main text figures.
- 3) Figure formatting: Scale bars must be present on all microscopy images, including inset magnifications. Molecular weight or nucleic acid size markers must be included on all gel electrophoresis. Aspect ratios of images may not be altered.
- 4) Statistical analysis: Error bars on graphic representations of numerical data must be clearly described in the figure legend. The number of independent data points (n) represented in a graph must be indicated in the legend. Statistical methods should be explained in full in the materials and methods. For figures presenting pooled data the statistical measure should be defined in the figure legends. Please also be sure to indicate the statistical tests used in each of your experiments (either in the figure legend itself or in a separate methods section) as well as the parameters of the test (for example, if you ran a t-test, please indicate if it was one- or two-sided, etc.). Also, if you used parametric tests, please indicate if the data distribution was tested for normality (and if so, how). If not, you must state something to the effect that "Data distribution was assumed to be normal but this was not formally tested."
- 5) Abstract and title: The abstract should be no longer than 160 words and should communicate the significance of the paper for a general audience. The title should be less than 100 characters including spaces. Make the title concise but accessible to a general readership.
- 6) Materials and methods: Should be comprehensive and not simply reference a previous publication for details on how an experiment was performed. Please provide full descriptions in the text for readers who may not have access to referenced manuscripts.
- 7) All antibodies, cell lines, animals, and tools used in the manuscript should be described in full, including accession numbers for materials available in a public repository such as the Resource Identification Portal. Please be sure to provide the sequences for all of your primers/oligos and RNAi constructs in the materials and methods. You must also indicate in the methods the source, species, and catalog numbers (where appropriate) for all of your antibodies. Please also indicate the acquisition and quantification methods for immunoblotting/western blots.
- 8) Microscope image acquisition: The following information must be provided about the acquisition and processing of images:
 - a. Make and model of microscope
 - b. Type, magnification, and numerical aperture of the objective lenses
 - c. Temperature
 - d. Imaging medium
 - e. Fluorochromes
 - f. Camera make and model
 - g. Acquisition software
 - h. Any software used for image processing subsequent to data acquisition. Please include details and types of operations involved (e.g., type of deconvolution, 3D reconstitutions, surface or volume rendering, gamma adjustments, etc.).

10) Supplemental materials: There are strict limits on the allowable amount of supplemental data. Articles may have up to 5 supplemental figures. Please also note that tables, like figures, should be provided as individual, editable files. A summary of all supplemental material should appear at the end of the Materials and methods section.

13) ORCID IDs: ORCID IDs are unique identifiers allowing researchers to create a record of their various scholarly contributions in a single place. Please note that ORCID IDs are now *required* for all authors. At resubmission of your final files, please be sure to provide your ORCID ID and those of all co-authors.

Please note that JCB now requires authors to submit Source Data used to generate figures containing gels and Western blots with all revised manuscripts. This Source Data consists of fully uncropped and unprocessed images for each gel/blot displayed in the main and supplemental figures. For assays performed using capillary electrophoresis and/or immunoassay-based detection, authors should instead provide the electropherogram graph(s) for each experiment, plotting fluorescence/chemiluminescence intensity vs. molecular weight/size. Please be sure to provide one Source Data file for each figure gels, blots, and/or capillary electrophoresis assays along with your revised manuscript files. File names for Source Data figures should be alphanumeric without any spaces or special characters (i.e., SourceDataF#, where F# refers to the associated main figure number or SourceDataFS# for those associated with Supplementary figures). For traditional gels and blots, the lanes of the gels/blots should be labeled as they are in the associated figure, the place where cropping was applied should be marked (with a box), and molecular weight/size standards should be labeled wherever possible. For capillary electrophoresis assays, each trace in the graph should be color-coded and labeled to indicate which protein, gene, or sample is being measured (please try to avoid red/green combinations to accommodate our color-blind readers).

Journal of Cell Biology now requires a data availability statement for all research article submissions. These statements will be published in the article directly above the Acknowledgments. The statement should address all data underlying the research presented in the manuscript. Please visit the JCB instructions for authors for guidelines and examples of statements at (<https://rupress.org/jcb/pages/editorial-policies#data-availability-statement>).

B. FINAL FILES:

****It is JCB policy that if requested, original data images must be made available to the editors. Failure to provide original images upon request will result in unavoidable delays in publication. Please ensure that you have access to all original data images prior to final submission.****

****The license to publish form must be signed before your manuscript can be sent to production. A link to the electronic license to publish form will be sent to the corresponding author only. Please take a moment to check your funder requirements before choosing the appropriate license.****

Thank you for your attention to these final processing requirements. Please revise and format the manuscript and upload materials within 7 days. If you need an extension for whatever reason, please let us know and we can work with you to determine a suitable revision period.

Thank you for this interesting contribution, we look forward to publishing your paper in Journal of Cell Biology.

Sincerely,

Johan Auwerx, MD, PhD
Monitoring Editor

Andrea L. Marat, PhD
Deputy Editor

Journal of Cell Biology

Reviewer #1 (Comments to the Authors (Required)):

The authors addressed my requests beyond my expectations; congratulations to them for this solid paper.

Reviewer #2 (Comments to the Authors (Required)):

The authors have satisfied my concerns. The paper is improved, and the data supporting the model are expansive. I particularly appreciate the efforts to include new proteomics data.